# A synthetic cell-free 36-enzyme reaction system for vitamin B$_{12}$ production

Qian Kang[1,2,3], Huan Fang [1,2,3], Mengjie Xiang[1,2], Kaixing Xiao[2,3], Pingtao Jiang [2,3], Chun You [1,2], Sang Yup Lee [4] ✉ & Dawei Zhang [1,2,3] ✉

Adenosylcobalamin (AdoCbl), a biologically active form of vitamin B$_{12}$ (coenzyme B$_{12}$), is one of the most complex metal-containing natural compounds and an essential vitamin for animals. However, AdoCbl can only be de novo synthesized by prokaryotes, and its industrial manufacturing to date was limited to bacterial fermentation. Here, we report a method for the synthesis of AdoCbl based on a cell-free reaction system performing a cascade of catalytic reactions from 5-aminolevulinic acid (5-ALA), an inexpensive compound. More than 30 biocatalytic reactions are integrated and optimized to achieve the complete cell-free synthesis of AdoCbl, after overcoming feedback inhibition, the complicated detection, instability of intermediate products, as well as imbalance and competition of cofactors. In the end, this cell-free system produces 417.41 µg/L and 5.78 mg/L of AdoCbl using 5-ALA and the purified intermediate product hydrogenobyrate as substrates, respectively. The strategies of coordinating synthetic modules of complex cell-free system describe here will be generally useful for developing cell-free platforms to produce complex natural compounds with long and complicated biosynthetic pathways.

Vitamin B$_{12}$ (cobalamin) plays important roles in cellular metabolism, especially in DNA synthesis, methylation, and mitochondrial metabolism[1], which is an essential vitamin composed of a cobalt-containing corrinoid ring, upper and lower ligands. The upper ligands of cobalamin can be 5'-deoxyadenosine, methyl, hydroxy, and cyano group, resulting in 5'-deoxyadenosylcobalamin (AdoCbl), methylcobalamin (MeCbl), hydroxocobalamin (OHCbl) and cyanocobalamin (CNCbl), respectively. MeCbl and AdoCbl are the natural bioactive forms of vitamin B$_{12}$, while CNCbl is an industrially produced stable form and OHCbl is a hydrate that results from breaking the Co−C bond in AdoCbl.

Woodward and Eschemoser, together with about 100 students and researchers, exerted 11 years of efforts to successfully develop a process for the complete chemosynthesis of vitamin B$_{12}$ in 1976[2]. Due

to the complex and difficult chemical synthetic process, however, vitamin B$_{12}$ production has relied on the fermentation of microorganisms such as *Pseudomonas denitrificans*, *Propionibacterium freudenreichii*, *Propionibacterium shermanii* and *Sinorhizobium meliloti*[3–5] up to now. Although these strains produce high titers of vitamin B$_{12}$, their long period fermentation process and limited genetic engineering tools hinder further improvement. In the search for alternative production methods, various key intermediates or derivatives, such as hydrogenobyrate (HBA)[6], hydrogenobyrinate *a, c-* diamide (HBAD)[7] and hydrogenobyric acid (Hby)[8], have been obtained from engineered *Eschericia coli* strains, from which the titer of HBAD and Hby were reported to be 2.94 mg/L and 2.95 mg/L, respectively. More recently, a metabolically engineered *E. coli* platform[9,10], which expressed over 30 heterologous enzymes, was able to produce vitamin

[1]University of Chinese Academy of Sciences, No.19 (A) Yuquan Road, Shijingshan District, 100049 Beijing, China. [2]Tianjin Institute of Industrial Biotechnology, Chinese Academy of Sciences, 32 Xi Qi Dao, Tianjin Airport Economic Area, 300308 Tianjin, China. [3]Key Laboratory of Engineering Biology for Low-Carbon Manufacturing, Tianjin Institute of Industrial Biotechnology, Chinese Academy of Sciences, 300308 Tianjin, China. [4]Department of Chemical and Biomolecular Engineering (BK21 four program), Korea Advanced Institute of Science and Technology (KAIST), 291 Daehak-ro, Yuseong-gu, Daejeon 34141, Republic of Korea. ✉e-mail: leesy@kaist.ac.kr; zhang_dw@tib.cas.cn

$B_{12}$ with a titer of 0.67 mg/L in 24 h of fermentation[10]. However, due to intricate and protracted biosynthetic pathway of vitamin $B_{12}$, further metabolic engineering to enhance production remains a significant challenge. Therefore, there has been a great demand for alternative method for vitamin $B_{12}$ synthesis that does not exclusively rely on chemocatalysis or bacteria fermentation, which led us to consider a cell-free platform.

A cell-free enzymatic system is an emerging alternative to metabolic engineering because cascade catalysis can provide distinct advantages compared with microbial fermentation. The reconstructed cell-free platform is composed of numerous enzymes and cofactors required to achieve complicated biotransformation. Compared with metabolic engineering, cell-free synthesis is advantageous in some cases due to flexible engineering, efficient separation and purification of intermediates and the product, high tolerance to toxic intermediate compounds, and high titer, yield and productivity[11,12].

Recently, cell-free synthesis of target compounds has been increasingly pursued. For example, the fatty acid biosynthesis pathway from malonyl-CoA and acetyl-CoA was reconstituted and optimized in vitro using nine distinct native enzymes of *E. coli* and NAD(P)H[13]. Furthermore, the synthetic pathway was extended with a four-subunit ATP-dependent acetyl-CoA carboxylase to control the percentage of unsaturated fatty acids between 10% and 50%[14]. Moreover, a reconstituted synthetic system using glucose as a substrate allowed the production of free fatty acids to reach >9% of the theoretical yield[15]. In another study, cell-free synthesis of monoterpenes from glucose was implemented using 27 enzymes, which led to the production of 15 g/L of sabinene with a conversion yield of 95%[16]. Also, cell-free enzymatic systems were reconstructed to produce cannabinoids[17,18]. More recently, artificial synthesis of starch from $CO_2$ catalyzed by a cell-free pathway was realized, marking great progress toward the biomanufacturing of starch without relying on plant sources[19].

However, the cell-free synthesis of cobalamin has not yet been realized due to the long synthetic pathway, and only a handful of intermediates synthesized by enzymatic cascades have been reported. For example, the metal-free corrinoid precursor HBA was synthesized from 5-aminolevulinic acid (5-ALA) by 17 enzyme-catalyzed reactions in a single reaction vessel[20,21]. Multienzyme catalysis of nucleotide loop assembly (NLA), assembling the lower ligand from 5,6-dimethyl benzimidazole (DMBI) to adenosylcobinamide to produce AdoCbl, was reconstituted in vitro by combining crude cell extracts and purified enzymes[22]. Furthermore, cell-free synthesis of precorrin-2, the key precursor of cobalamin and siroheme, was carried out using a cell-free four-enzyme reaction system, which was optimized using response surface methodology to increase the productivity of precorrin-2 to 0.1950 μM/min[23]. However, none of these earlier studies resulted in the de novo synthesis of cobalamin due to the complex pathway with dozens of reaction steps and compounds.

In this study, we successfully reconstituted a synthetic cell-free platform to synthesize AdoCbl using 5-ALA as a substrate. To achieve this, the whole synthetic pathway, comprising 32 overall steps catalyzed by 36 enzymes, including 8 regeneration reactions catalyzed by 10 additional enzymes in the regeneration modules, from ten different microorganisms, was implemented as a synthetic AdoCbl production system. The entire pathway was separated into five synthetic modules and five cofactor regeneration modules. A design-test-optimize cycle was performed separately for every synthetic module. After relieving the oxidation of intermediate uroporphyrinogen III and strong inhibition of methyltransferase caused by S-adenosylhomocysteine (SAH), the titer of key intermediate HBA approached 10.23 mg/L through 12 h cell-free synthetic reaction. Also, we developed a method for stop-flow detection and direct identification by LC-MS of adenosylcobyrate (AdoCby) in cell-free cascade reactions. Furthermore, the customized cofactor regeneration modules for the complete vitamin $B_{12}$ synthetic pathway were designed and the supply of 5-ALA, SAM, NADH, ATP, and

glutamine was optimized, while reducing the accumulation of SAH and polyphosphate. Finally, the whole synthetic system was reconstituted and used to produce 417.42 μg/L of AdoCbl from 5-ALA. To examine the application potential, the cell-free reaction system was optimized for the production of AdoCbl from HBA, and the final AdoCbl titer reached 5.78 mg/L in 14 h, showing a significant increase compared with the initial reaction system and surpassing what was achieved using the reconstituted *E. coli*[10].

## Results

### Design of a cell-free platform for the synthesis of AdoCbl

Two metabolic pathways have so far been known for the biosynthesis of AdoCbl, classified according to the order of cobalt insertion and methylation of the corrin ring, referred to as the early cobalt insertion route (anaerobic pathway) and late cobalt insertion route (aerobic pathway). A schematic representation of our cell-free synthetic approach based on the aerobic pathway is outlined in Fig. 1. In consideration of thermodynamic characteristics and detection of intermediate products, the whole synthetic pathway was separated into five modules that were implemented and optimized individually. Precorrin-2 is synthesized by HemB, HemC, HemD, and CobA in the precursor module using 5-ALA as a substrate, resulting in the assembly of the central porphyrin ring. The stoichiometric proportion of substrate: product in the precursor module is 8:1. The HBA module is responsible for a series of modifications of the porphyrin ring precursor to yield HBA, including six S-adenosylmethionine (SAM)-depended methylation reactions at $C^{20}$, $C^{17}$, $C^{11}$, $C^1$, $C^{15}$, and $C^5$ of the porphyrin ring and ring contraction, respectively. Further amidation and upper ligand assembly are carried out in the AdoCby module, which includes the amidation, cobalt chelation reaction, cobalt reduction and adenylation reaction, leading to the production of AdoCby from HBA. Then, the nucleotide loop from 5,6-dimethyl benzimidazole (DMBI) is assembled as a ligand of AdoCby in the AdoCbl module, resulting in the synthesis of AdoCbl. In addition, (R)−1-amino-2-propanol-O-2-P and α-ribazole-5′-P are synthesized in the branch module-1 and branch module-2, which are provided as substrates to the AdoCby and AdoCbl modules, respectively. To achieve this, 24 reactions catalyzed by 26 synthetic enzymes from different microorganisms were chosen and assembled. The enzyme classification numbers (EC number) with relevant information including the source organism, thermodynamic characteristics (the standard Gibbs free energy change, $\Delta G'^\circ$) and specific bioreactions catalyzed by every enzyme are listed in Supplementary Table 1. The $\Delta G'^\circ$ of each enzymatic reaction was computed using equilibrator (https://equilibrator.weizmann.ac.il/) at pH 8.0 and ionic strength of 0.1 M, or obtained from MetaCyc (https://metacyc.org/), and is shown in Fig. 2. Although reactions synthesizing HBA, CBAD, and AdoCBAD are thermodynamically unfavorable, the complete pathway has a net $\Delta G'^\circ$ of −486.80 kcal/mol, and is thus thermodynamically favorable.

### Optimization of the precursor synthesis module

Eight 5-ALA molecules are assembled into one precorrin-2 catalyzed by porphobilinogen synthase (HemB), porphobilinogen deaminase (HemC), uroporphyrinogen-III synthase (HemD), and uroporphyrinogen-III $^{(C2,7)}$-methyltransferase (CobA) in the precursor synthesis module (Fig. 3a). However, an unknown reddish-brown intermediate with high absorbance at 400 nm was detected in the HBA synthesis system and considered formed in the precursor module (Supplementary Fig. 1a). After identification by LC-MS, the reddish-brown by-product was confirmed to be uroporphyrin III (Uro III), the oxidized form of uroporphyrinogen III (Urogen III) and a dead-end by-product (Fig. 3b). In order to confirm this putative oxidation reaction, a reference standard of Urogen III was synthesized in an anaerobic chamber by cascade catalysis and quantified based on a previous report[23,24]. The results indicated that 83.3% of Urogen III was oxidized after 7 h of exposure to air at 30 °C,

suggesting the high oxygen sensitivity of Urogen III (Supplementary Fig. 1b). To decrease the oxidation of Urogen III, different reductants were added to the precursor synthetic reaction system, and their effectiveness was assessed by detecting Uro III and sirohydrochlorin, a downstream fluorescent derivative of precorrin-2 produced by SirC[23]. The addition of 2% (v/v) β-mercaptoethanol exerted the best effect, preventing the oxidation of 80.7 ± 1% Urogen III, and accordingly increasing the titer of sirohydrochlorin by 285.5 ± 3% (Supplementary Fig. 1c, d), which solved the bottleneck in the precursor synthesis module.

## Synthesis of HBA in a cell-free reaction system

Precorrin-2 synthesized above needs to be further converted to HBA by precorrin-2(C20)-methyltransferase (CobI), precorrin 3B synthase (CobG), precorrin-3B(C17)-methyltransferase (CobJ), precorrin-4(C11)-methyltransferase (CobM), precorrin-5(C1)-methyltransferase (CobF), precorrin 6A reductase (CobK), precorrin-6B(C5, 15)-methyltransferase (CobL), and precorrin-8X methyl-mutase (CobH). To achieve this, the HBA1 strain. which is the *E. coli* MG1655 (DE3) strain harboring a pET28a-derived plasmid with an artificial cob operon encoding eight Cob genes from CobI to CobH (CobI, J, M, F, K, L, H from *R. capsulatus* SB 1003 genome and CobG from *B. melitensis* bv.1 str. 16 M genome), was constructed and employed as a source of crude cell extract for the synthesis of HBA. Since endogenous CysG of *E. coli* can convert Urogen III to precorrin-2 (in the reaction catalyzed by a C-terminal partial enzyme named CysGA) or siroheme (by the N-terminal partial enzyme named CysGB)[25,26], different strains derived from HBA1 with complete or partial knockout of the *cysG* operon were constructed used as the

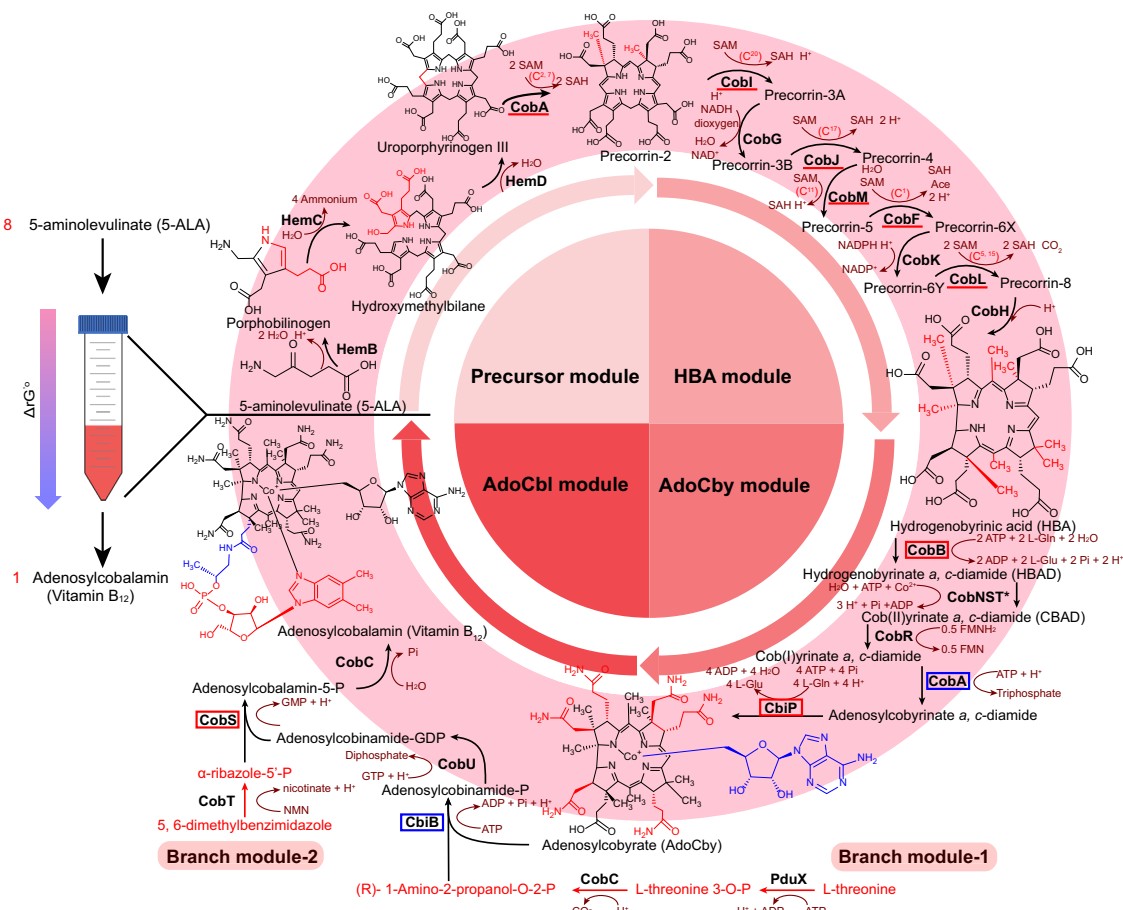

**Fig. 1 | The cell-free synthetic system for adenosylcobalamin production.**
Schematic diagram of AdoCbl synthetic pathway. The arrow with gradient color form red to blue in the left of illustration means that the Gibbs free energy of the whole catalytic pathway is decreased through the catalytic process from 5-ALA to Adenosylcobalamin. The whole pathway was separated into five modules, with compounds shown in black and enzymes in bold black fonts. Cofactors and by-products are shown in red brown. In the precursor module (upper left of the circle), eight 5-ALA units are assembled into one precorrin-2 catalyzed by HemB, HemC, and HemD, followed by modification by CobA. The red portion of the structural formula indicates the reaction catalyzed by the previous enzymatic reaction. In the HBA module (upper right of the circle), precorrin-2 is catalyzed by eight Cob enzymes to produce HBA, including methylation by CobI, CobJ, CobM, CobF, CobL with SAM serving as the methyl donors. The methyltransferase is highlighted with a red underline in HBA synthesis, and the methylation position is labeled above the reactions. The red portions in HBA structural formula indicate the eight methylated modifications catalyzed by the HBA module (and CobA in precursor module). In the AdoCby module (bottom right of the circle), HBA undergoes amidation, reduction

and adenylation, after which the adenosine group is assembled onto the porphyrin ring as the upper ligand to yield adenosylcobyrate (AdoCby). The six amidate groups introduced by CobB and CbiP are highlighted in red in the structure formula, while the adenosyl group introduced by CobA is highlighted in blue, and enzymes are marked with the same color frames as structures. The enzymes CobNST asterisked means the enzyme complex assembled with CobN, CobS and CobT. Two branch modules (red words in the bottom of the figure) are responsible for providing (R)−1-amino-2-propanol-O-2-P and α-ribazole-5′-P from L-threonine and 5, 6-dimetbenzimidazole in branch module-1 and -2, respectively. In the AdoCbl module (bottom left of the circle), the modified down ligand from 5,6-dimethylbenzimidazole is assembled onto adenosylcobyrate to form adenosylcobalamin. The highlighted enzymes are shown with blue or red frames with the same color indicating the structure parts introduced from the branch module, respectively. 5-ALA 5-aminolevulinate, HBA hydrogenobyrinic acid, HBAD hydrogenobyrinate *a, c*-diamide, CBAD cob(II)yrinate *a, c*-diamide, AdoCby adenosylcobyrate, AdoCbl, adenosylcobalamin, SAM S-adenosyl-L-methionine, SAH S-adenosyl-L-homocysteine.

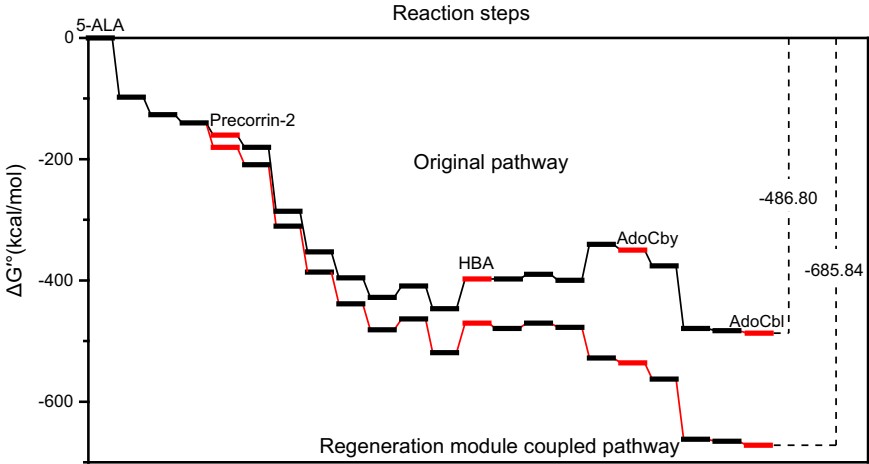

**Fig. 2 | The standard Gibbs free energy change in the whole AdoCbl synthesis pathway.** The standard Gibbs free energy change ($\Delta G'^{\circ}$) of every enzyme involved in our reaction system were listed in supplementary Table 1. The black line indicates the synthetic pathway of AdoCbl from 5-ALA, while the red line indicates the regeneration module coupled synthetic pathway which involved with MetK, MtnN, RocG, OGDH, HemA, GlnA, and PpA, which were calculated according to the consumption of SAM, NADH and L-glutamine and accumulated polyphosphate. Every bold platform in this line graph indicates an intermediate product, while the red bold lines highlight the key compounds at each end of a synthetic module. The $\Delta G'^{\circ}$ from 5-ALA to adenosylcobalamin is −486.80 kcal/mol and −685.84 kcal/mol in original pathway and regeneration module coupled pathway, respectively. 5-ALA 5-aminolevulinate, HBA hydrogenobyrinic acid, AdoCby adenosylcobyrate, AdoCbl adenosylcobalamin. Source data are provided as a Source Data file.

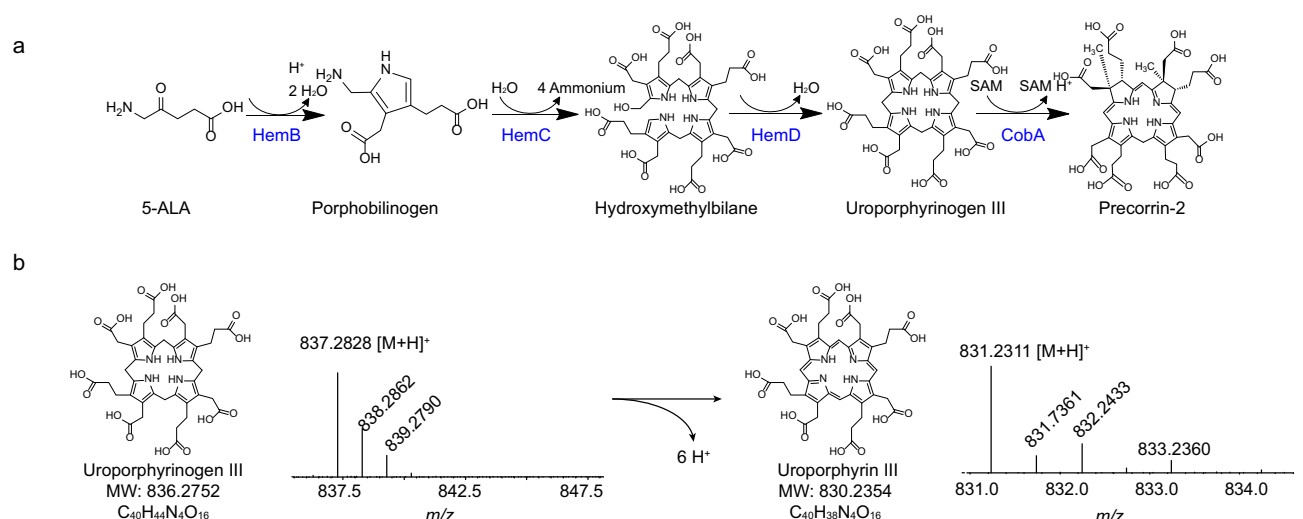

**Fig. 3 | Dead-end by-product uroporphyrin III in the precursor module. a** Schematic diagram of cascade reaction in the precursor module. **b** illustration and LC-MS detection of uroporphyrinogen III oxidating. 5-ALA 5-aminolevulinate. Source data are provided as a Source Data file.

sources of crude cell extracts for the HBA module: HBA1, MG1655 (DE3)-HBA; HBA2, MG1655 ΔCysG (DE3)-HBA; HBA3, *E. coli* MG1655 ΔCysGA (DE3)-HBA; HBA4, MG1655 ΔCysGB (DE3)-HBA. HBA was synthesized from 5-ALA by combining the crude cell extracts containing the HBA pathway and purified precursor module enzymes, after which the product was detected by LC-MS and quantified by HPLC (Fig. 4a). Through fed-batch addition 1 mM 5-ALA and SAM every hour, the synthetic system composed of the HBA2 crude cell extract produced a maximum HBA titer of 8.40 mg/L in 5 h with a productivity of 1.68 mg/L/h (Fig. 4b). This result suggests that blocking the endogenous branch pathway from precorrin-2 to siroheme in HBA crude cell extract had a positive effect on the synthesis of HBA.

In the HBA synthetic reaction system described above, SAM was added in a fed-batch mode of 1 mM every hour because it was reported to be prone to depurination and epimerization at the sulfur center under physiological condition[27]. Furthermore, accumulated feedback inhibition of methyltransferases by SAH was considered to be a serious potential barrier to multi-transmethylation HBA synthetic pathway. To solved the SAM derived puzzle, the cascade reaction of the precursor module, including the methyl transfer reaction catalyzed by CobA, was used to test the inhibition related to SAH. In the process of precursor module test, there was no obvious increase of the precorrin-2 titer after supplementation of SAM exceed 5 mM. Moreover, the boost in precorrin-2 production was not in proportion to the increase in SAM input, which suggests a possible blockage in the methylation process (Supplementary Fig. 2b). In addition, the inhibition of CobA by SAH was confirmed by adding extra SAH, and addition of 0.01 mM SAH significantly reduced the preocrrin-2 titer (Supplementary Fig. 2c). To relieve the inhibition, MtnN was employed to remove SAH from the reaction system by depurination (illustrated in Supplementary Fig. 2a). However, adding MtnN alone did not relieve the strong inhibition when excess SAM (more than 3 mM in our testing reaction system) was introduced (Supplementary Fig. 2d). For further optimization, methionine adenosyltransferase MetK was employed to produce SAM

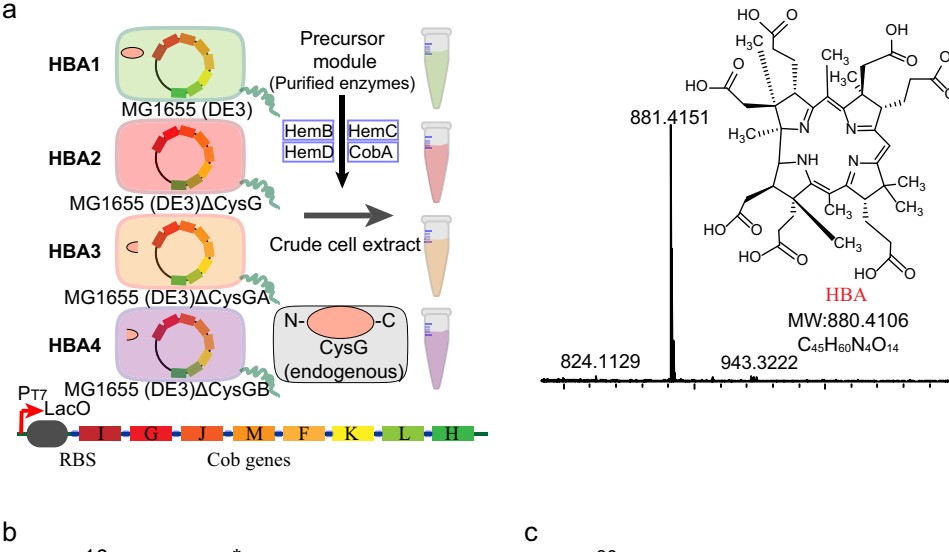

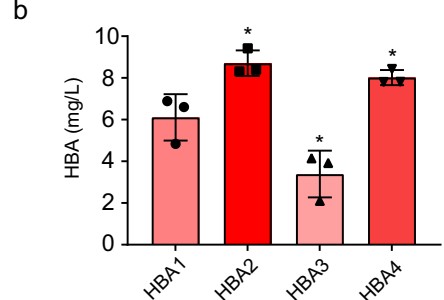

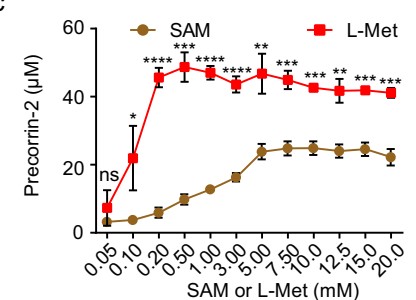

**Fig. 4 | Cell-free synthesis in the HBA module. a** Schematic diagram of the cell free synthesis of HBA and LC-MS detection of synthetic HBA. Four different engineered *E. coli* MG1655 (DE3) strains harboring one pET28a-drived plasmid with the HBA *Cob* operon, working as a source of crude cell extracts to synthesize HBA. HemB, HemC, HemD, and CobA enzymes used in the form of purified enzymes, but other Cob enzymes was expressed in a pET28a-drived plasmid illustrated in graph and used in crude cell extracts. The knock out of endogenous complete or partial CysG are also illustrated in this graph. **b** Synthesis of HBA by different crude cell extracts supplied Cob enzymes. Every reaction mixture contained 0.1 μM HemB, 1 μM HemC, 1 μM HemD, 10 μM CobA, 77.9 mg/ml wet cell weight of HBA crude cell extract (corresponding to 8.9 mg/ml dry cell weight of HBA crude cell extract and 3.1 mg/ml total protein of HBA crude cell extract. Detailed information was illustrated in Supplementary Fig. 11), 2 mM NADH, 1 mM NADPH, 5 mM MgCl₂, 10 mM KCl, 5 mM NaCl in 50 mM Tris-HCl (pH 8.0) buffer, 5 mM 5-ALA and 5 mM SAM was added 1 mM per hour for 5 h, and synthetic HBA was treated and detected by HPLC according to detection method. Reactions were performed in triplicate (*n* = 3 biologically independent samples) and data are presented as mean values ± SD. Two-sided unpaired

*t* test is carried out between HBA2, 3, 4 reactions and HBA1. Unpaired t test of data: HBA2 to HBA1, *P* = 0.0240 (*t* = 3.540); HBA3 to HBA1, *P* = 0.0409 (*t* = 2.977); HBA4 to HBA1, *P* = 0.0476 (*t* = 2.825). *, *P* < 0.05. **c** Optimization of the cascade reactions in the precursor module by introducing MetK and MtnN. The reaction was performed in 100 mM Tris-HCl buffer (pH 8.0) with 5 mM MgCl₂, 100 mM KCl, 50 mM NaCl, 0.1 μM HemB, 1 μM HemC, 1 μM HemD, 10 μM CobA and 3 mM ALA. 5 μM MetK, 5 μM PpK, 10 μM MtnN, 2 mM AMP, and 1 mM SMPP were externally added to the L-Met reaction mixture. Titration of SAM and L-Met. Reactions were performed in triplicate (*n* = 3 biologically independent samples) and data are presented as mean values ± SD. Two-sided unpaired *t* test is carried out with the titer of precorrin-2 between using L-Met and SAM in the same concentration. Unpaired *t* test of data: 0.05, *P* = 0.2468 (*t* = 1.355); 0.1, *P* = 0.0296 (*t* = 3.313); 0.2, *P* < 0.0001 (*t* = 221.37); 0.5, *P* = 0.0001 (*t* = 14.71); 1, *P* < 0.0001 (*t* = 29.37); 3, *P* < 0.0001 (*t* = 17.62); 5, *P* = 0.0032 (*t* = 6.300); 7.5, *P* = 0.0005 (*t* = 10.23); 10, *P* = 0.0001 (*t* = 14.06); 12.5, *P* = 0.0016 (*t* = 7.647); 15, *P* = 0.0001 (*t* = 14.78); 20, *P* = 0.0003 (*t* = 11.65). ns, *P* > 0.05; *, *P* < 0.05; **, *P* < 0.01; ***, *P* < 0.001; ****, *P* < 0.0001. Source data are provided as a Source Data file.

from L-methionine (L-Met) and consequently to avoid addition of SAM. This approach of substituting SAM with the equimolar amount of L-Met as a methyl donor increased the titer of precorrin-2 (Supplementary Fig. 2e). After introducing SAM module contained of MetK and MtnN in the precursor synthesis module, both the titer and yield of precorrin-2 were increased and uroporphyrinogen III accumulation was decreased (Supplementary Fig. 2f, Fig. 4c).

To exploring the suitable buffer and pH value condition of mixed multienzymes reaction system[28–30], Tris-HCl and Hepes-NaOH buffers with different pH values were examined in the reaction system. The HBA titer increased at higher pH values, and Hepes-NaOH was a better buffer for HBA production (Fig. 5a). Taken together, after combining the optimized SAM module and reaction environment, the HBA titer increased to 8.65 mg/L in one-pot substrate addition mode, which was much higher than that (2.00 mg/L) obtained with the original one-pot reaction system, and was similar to the titer initially obtained in a fed-batch substrate addition mode (Fig. 5b). After that, when the cascade

reaction was strengthened by increasing crude cell extract and applying fed-batch 5-ALA addition 0.3 mM every 2 h in 10 h reaction period, the HBA titer increased from 10.23 to 10.95 mg/L, indicating that the 5-ALA addition mode does not alter the overall reaction in HBA synthesis system (Fig. 5c). Taking into account the unfavorable thermodynamic preference of the CobH reaction to form HBA from precorrin-8 (Fig. 2), we performed reactions from 5-ALA to synthesize HBAD, with the aim of driving metabolic flux towards the desired product. Unexpectedly, HBAD was hardly detected and titer of HBA is barely increased (Supplementary Fig. 3). This was confirmed to be caused by the driving force of cascade reaction and reaction equilibrium of CobB reaction, which will be described in the following chapter.

### Synthesis of AdoCby from HBA

In the AdoCby module, HBA is amidated in the a and c side chains of corrin ring catalyzed by CobB, and then one Co²⁺ is inserted in the

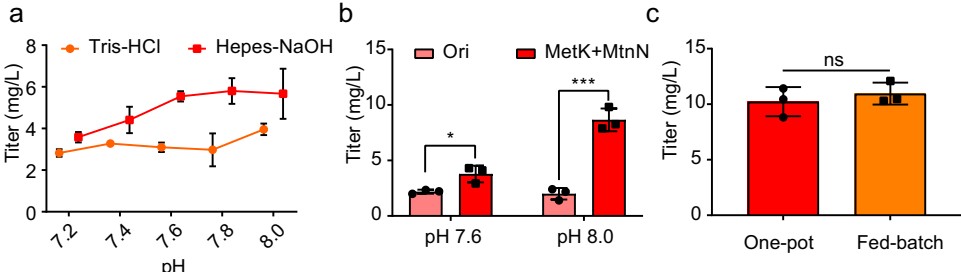

**Fig. 5 | Optimization of the HBA synthetic module. a** Buffer screening of the HBA synthetic module with different pH value. Reactions were performed in triplicate ($n = 3$ biologically independent samples) and shown with line running through the mean values ± SD. **b** Evaluating the optimization of the sources of SAM module and reaction buffer. Reactions were performed in triplicate ($n = 3$ biologically independent samples) and data are presented as mean values ± SD. Two-sided unpaired $t$ test is carried out with titer in different reaction system in the same pH condition. Unpaired t test of data: pH 7.6, $P = 0.0226$ ($t = 3.606$); pH 8.0, $P = 0.0006$ ($t = 10.02$). \*, $P < 0.05$; \*\*\*, $P < 0.001$. **c** Influence of 5-ALA addition mode in the optimized HBA reaction system. Reactions were performed in triplicate ($n = 3$ biologically independent samples) and data are presented as mean values ± SD. Two-sided unpaired $t$ test is carried out with titer between one-pot reaction and fed-batch reaction. Unpaired $t$ test: $P = 0.4869$ ($t = 0.7650$). ns, $P > 0.05$. Source data are provided as a Source Data file.

center of the corrin ring by the CobNST enzyme complex. The inserted $Co^{2+}$ was further reduced to $Co^{1+}$ by CobR, and linked to an adenosyl group from ATP catalyzed by CobA. In the end of this synthetic module, the b, d, e, g side chains of corrin ring are amidated by CbiP to yield AdoCby (Fig. 6a).

In AdoCby module, HBA is first catalyzed by hydrogenobyrinate a, c-diamide synthase to form hydrogenobyrinate a, c-diamide (HBAD). As a result of detecting HBA accumulation in an engineered E. coli strain producing vitamin B$_{12}$ in previous pre-experiment in vivo, it was hypothesized that the reaction synthesizing HBAD from HBA was a bottleneck reaction during the initial period of setting up the cell-free reaction system. In order to demonstrate this catalytic reaction and to screen for better hydrogenobyrinate a, c-diamide synthases, we purified the HBA standard using a previously reported enzyme-trap method[6]. Hydrogenobyrinate a, c-diamide synthase CobB from *Rhodobacter sphaeroides*, *Rhodobacter capsulatus*, *P. denitrificans*, *S. meliloti*, and *Brucella melitensis*, as well as cobyrinate a, c-diamide synthase CbiA from *Salmonella typhimurium*, *Bacillus megaterium*, and *Methanococcus jannaschii* were all reported could convert HBA into HBAD in natural metabolic pathway (but CbiA enzymes prefer to catalyze the derivant of HBA, cobyrinate, to form cobyrinate a, c-diamide in natural anaerobiotic metabolic pathway), by catalyzing the amidation at the a and c side chains of the corrin ring using L-glutamine as amide donor[31–38]. Enzymes above were chosen from natural organisms with higher titers of vitamin B$_{12}$, according to the conservation of substrate binding sites and active catalytic residues based on BLAST analysis (Supplementary Fig. 4). After detecting the transformation yield according to a previously reported HPLC method[10], CobB from R. capsulatus (RcCobB) was considered to have the highest conversion yield, which reached practically 100% showing as no detectable HBA left in the reaction system (Fig. 6c). However, different amounts of the intermediate hydrogenobyrinate c-monoamide (HBAM) were monitored by LC-MS in every reaction system, including the one catalyzed by RcCobB (Fig. 6b, d), which could explain the stoichiometric imbalance of HBA and HBAD in every reactant. Nevertheless, it is hypothesized that the accumulation of HBAM was caused by the reaction equilibrium of CobB ($\Delta G'^{\circ} = -0.56$ kcal/mol) and could be reduced by the driving force of downstream reactions in the metabolic pathway.

In the aerobic AdoCbl synthesis pathway, cobalt is inserted into HBAD catalyzed by the cobaltochelatase complex composed of CobN, CobS and CobT[39]. A cascade reaction catalyzed by CobB and CobNST (CobN from *B. melitensis* and CobS, CobT from *S. meliloti*, selected based on a reference from a previous reported article[10]) was preformed and evaluated by monitoring the consumption of substrate HBA, the accumulation of intermediates HBAM and HBAD, as well as the calculated amount of synthetic CBAD. The titration results showed

that a suitable amount of ATP is crucial to the yield of this cascade reaction. Insufficient concentrations of ATP are insufficient to initiate the cascade reactions, but excessive addition of ATP results in a negative impact on the product yield. This is hypothesized to be due to the precipitation of enzymes after introducing ATP and possibly caused by the slightly change of pH immediately after adding ATP. In addition, ATP regeneration, which will be further described later, can reinforce the cascade reactions when the initial ATP input is low by the regeneration of ATP from polyphosphate (Supplementary Fig. 5b). Furthermore, high concentrations of $Co^{2+}$ also decreased the yield of this system (Supplementary Fig. 5a), which was likely due to negative effects of $Co^{2+}$ on the enzymes. Moreover, the driving force of CobNST reaction to the equilibrium of CobB reaction was also proved. This suggests that the CobB reaction reaches an equilibrium state between HBA, HBAM and HBAD, resulting in an incomplete reaction. However, the introduction of CobNST reaction leads to a redirection of the metabolic flux towards the formation of CBAD and significantly reduction in the accumulation of HBAM and HBAD. In this cascade reaction, the highest yield of CBAD from HBA reached 92.40%, and there was also a considerable reduction in the accumulation of HBAM (Supplementary Fig. 5c), which also explains the lack of HBAD acquisition in the HBAD-stop reaction system in Supplementary Fig. 3.

It has been reported that the reduction of the central $Co^{2+}$ of CBAD to $Co^{1+}$ is a prerequisite for attachment of an adenosyl group to AdoCby from ATP catalyzed by adenosyltransferase, working as the upper axial ligand of AdoCby and derivant[40]. There are at least two distinct ways in nature to reduce the central cobalt in the corrin ring, in which one is catalyzed by ferredoxin (flavodoxin): NADH reductase and flavodoxin A proteins, while the other is catalyzed by CBAD reductase[40]. CBAD reductase CobR from *B. melitensis* was chosen to catalyze the reduction of the central $Co^{2+}$ in CBAD in our cell-free synthetic system. Purified CobR appeared bright yellow and exhibited a similar UV-visible absorbance spectrum to flavin nucleotide (Supplementary Fig. 6a), indicating spontaneous binding of flavin mononucleotide (FMN) and flavin adenine dinucleotide (FAD) during heterologous overexpression[40]. NADH-flavin reductase Fre from E. coli was introduced to catalyze FMN to FMNH$_2$, supplying to the CobR reaction. However, we unexpectedly found that CobR can be bleached by NADH solely, and CobR even can catalyze the reduction of FMN in the exist of NADH, which meaning that CobR have the activity of NADH-depended flavin nucleotide reductase. We therefore hypothesized that CobR can separately catalyze a cofactor (such as FMN and FAD) and CBAD using NADH directly. To further explore the catalytic process, the redox status of FMN was monitored in the mixture including FMN-NADH-CobR, FMN-NADH-Fre, FMN-NADH-CobR-Fre (FMN-reductase from *E. coli*), or FMN-NADH-CobR-Fre-CobA

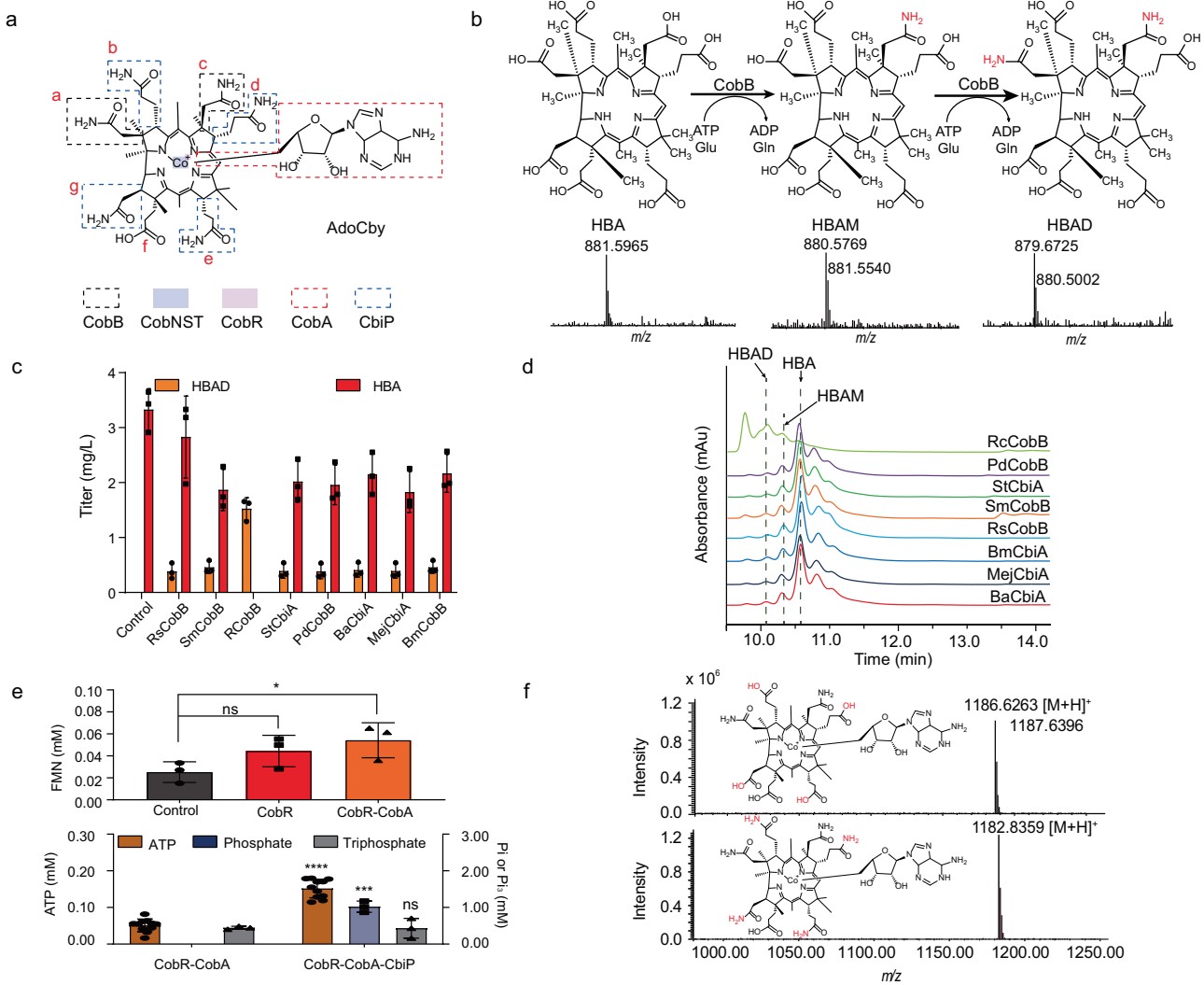

**Fig. 6 | Reactions in the AdoCby module. a** General schematic diagram of reactions in AdoCby synthetic module. **b** Schematic diagram of reaction catalyzed by hydrogenobyrinate *a, c*-diamide synthase and corresponding UPLC-MS detection. **c** Screening of eight hydrogenobyrinate *a, c*- diamide synthases from different organisms. Reactions were performed in triplicate ($n = 3$ biologically independent samples) and data are presented as mean values ± SD. **d** Chromatogram of the screened hydrogenobyrinate *a, c*-diamide synthases reaction. The retention time of HBA was 10.603 min, that of HBAM was 10.426 min and that of HBAD was 10.168 min. **e** Upper graph, detecting the CBAD reduction catalyzed by CobR. Control: FMN, NADH and CobR were incubated in 100 mM Tris·HCl buffer (pH 8.0); CobR: Control group with CBAD reactant (HBAD incubated after cobalt chelation); CobR-CobA: CobR reaction system with external addition of cobinamide adenosyltransferase CobA. Reactions above were incubated at 32 °C for 6 min, and the increasing concentration of FMN demonstrating the reduction of CBAD. Reactions were performed in triplicate ($n = 3$ biologically independent samples) and data are presented as mean values ± SD. Two-sided unpaired *t* test is carried out with FMN recovery between

CobR reaction or CobR-CobA reaction with control reaction. Unpaired *t* test: CobR to control, $P = 0.1246$ ($t = 1.939$); CobR-CobA to control, $P = 0.0157$ ($t = 4.033$). ns, $P > 0.05$; \*, $P < 0.05$. Lower graph, Stop-flow detection in AdoCby synthetic module. CobR-CobA, cascade reaction system from HBAD to adenosylcobyrinate *a, c*-diamide. CobR-CobA-CbiP, cascade reaction system from HBAD to AdoCby. The results were normalized to the control. Reactions were performed in triplicate ($n = 3$ biologically independent samples) with detection of Pi and in decuple ($n = 10$ biologically independent samples) with detection of ATP. Data are presented as mean values ± SD. Two-sided unpaired *t* test is carried out with ATP, phosphate and triphosphate between CobR-CobA reaction and CobR-CobA-CbiP reaction. Unpaired *t* test: ATP, $P < 0.0001$ ($t = 10.91$); phosphate, $P = 0.0003$ ($t = 11.54$); triphosphate, $P = 0.9345$ ($t = 0.08752$). ns, $P > 0.05$; \*\*\*, $P < 0.001$; \*\*\*\*, $P < 0.0001$. **f** UPLC-MS detection of synthetic adenosylcobyrinate *a, c*-diamide and adenosylcobyrate. HBA hydrogenobyrinic acid, HBAD hydrogenobyrate *a, c*- diamide, HBAM hydrogenobyrate *c*- monoamide, Pi phosphate, Pi₃ triphosphate. Source data are provided as a Source Data file.

(adenosyltransferase from *S. typhimurium*), in which FMN were immediately reduced to FMNH₂ in each reactant (Supplementary Fig. 6b), confirming the NADH-FMN reductase activity of CobR. Enzyme kinetic parameter of CobR was also measured in the reduction of CobR. The Km for FMN was $0.17 \pm 0.03$ mM and the $V_{max}$ was $64.91 \pm 4.20$ mM min⁻¹ mg⁻¹, corresponding to a $K_{cat}$ of $60.10 \pm 3.89$ s⁻¹ (Supplementary Fig. 6c), indicating that CobR can efficiently catalyze the reduction of FMN. In addition, we attempted to split the two reduction steps of FMN and CBAD in CobR catalysis. In the process of catalysis, FMN was efficiently converted by CobR to FMNH₂ using NADH, after which FMNH₂ was spontaneously oxidized in

contact with atmospheric oxygen. When CBAD was added, FMN was gradually regenerated by CobR through the FMNH₂-dependent reduction of CBAD to form cob(I)yrinic acid *a, c*-diamide (CBAD(I)) (Supplementary Fig. 6d). In addition, the CobR-CobA reaction mixture could more efficiently reduced CBAD, corresponding to that CobA was reported to interact with CobR to overcome the large thermodynamic barrier required for Co²⁺ reduction[40].

Because the intermediate of the synthetic pathway from CBAD to AdoCby are challenging to detect directly, we demonstrated the cascade reactions catalyzed by CobR, CobA and adenosylcobyrate synthase CbiP by monitoring the cofactors and by-product such as FMN,

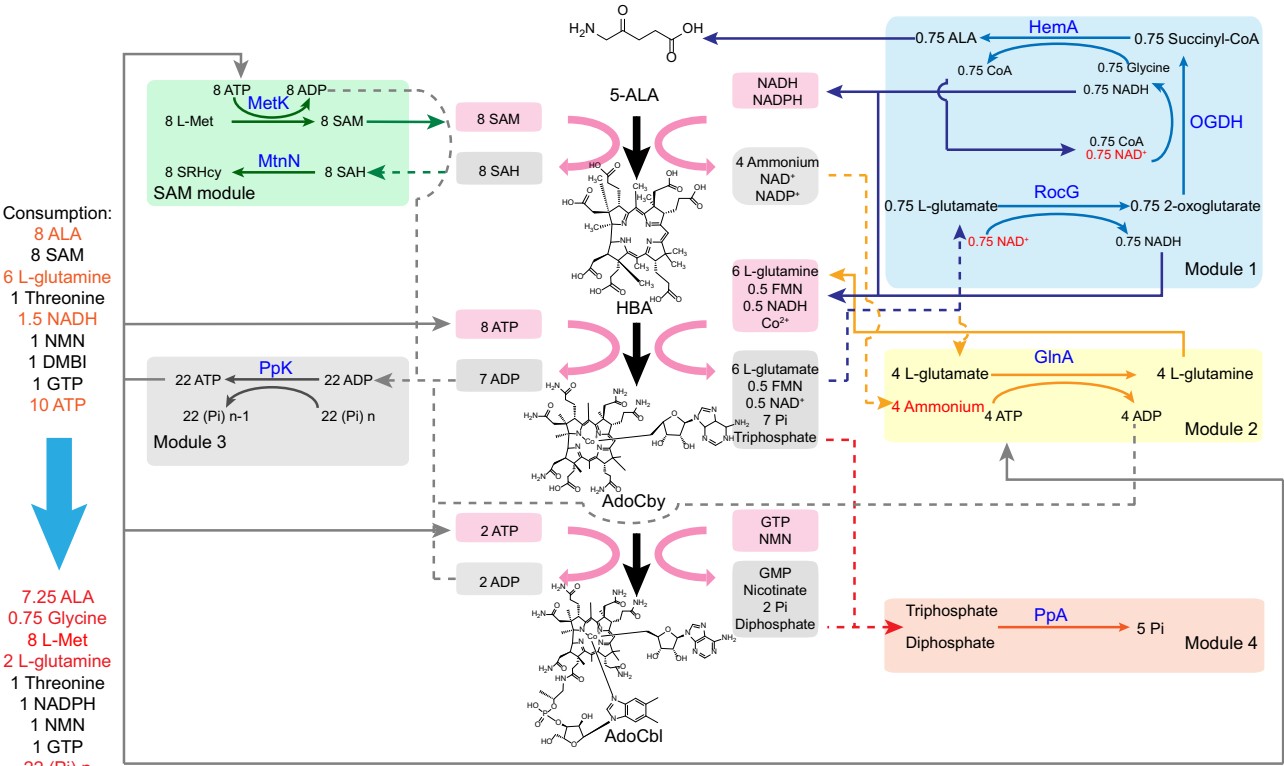

**Fig. 7 | Designed regeneration system for AdoCbl synthesis system.** The AdoCbl synthetic pathway is simplified into three key parts showing in the middle of this graph. Consumption (red box) and production (gray box) of each cofactor was listed around the reaction pathway. The regeneration module was split into five sub-modules shown in the two sides. In module 1, NADH and 5-ALA were regenerated from NAD⁺ and ʟ-glutamate catalyzed by RocG, OGDH and HemA. NAD⁺ (highlight in red words) was considered as the stoichiometric limiting reagent in the theoretical calculation in this graph. In module 2, ʟ-glutamine was regenerated from ʟ-glutamate by consuming same stoichiometric ATP. Ammonium (highlight in red words) was considered as the stoichiometric limiting reagent in the theoretical

calculation in this graph. In module 3, ATP was regenerated from ADP by PpK at the expense of externally added polyphosphate (Pi(n)). In module 4, accumulated triphosphate and diphosphate were hydrolyzed by PpA to prevent feedback inhibition. In SAM module, SAM was synthesized from ʟ-Met by MetK consuming ATP and by-product SAH was removed by MtnN to SRHcy. 5-ALA 5-aminolevulinate; HBA hydrogenobyrinic acid, AdoCby adenosylcobyrate, AdoCbl adenosylcobalamin, SAM S-adenosyl-ʟ-methionine, SAH S-adenosyl-ʟ-homocysteine, ʟ-Met ʟ-methionine, SRHcy S-ribosyl-homocysteine, NMN β-nicotinamide ᴅ-ribonucleotide, Pi phosphate.

ATP, triphosphate and phosphate. For detecting the reduction catalyzed by CobR, the separation strategy demonstrated above was carried out to detect the regeneration of FMN from FMNH₂ after the addition of CBAD. FMN was incubated with CobR, Fre, CobA, ATP and NADH at 32 °C for 3 min to generate FMNH₂, after which the CBAD reactant was introduced. FMN in the reaction system was recovered faster than the spontaneous recovery in the control group, which was calculated that 0.019 mM and 0.032 mM CBAD were respectively reduced to CBAD(I) in the CobR and CobR-CobA reaction systems in 6 min (Fig. 6e). Additionally, we also used the stop-flow method to detect the consumption of ATP and production of triphosphate and phosphate to evaluate the cascade reaction of CobA and CbiP in vitro. In the CBAD-CobR-CobA cascade catalysis, 0.44 mM triphosphate was detected, demonstrating the adenosyl transformation catalyzed by CobA. In the CBAD-CobR-CobA-CbiP cascade reaction, excess 1.02 mM phosphate was detected in the presence of CbiP compared to the CobA-stop reaction system, and 0.43 mM triphosphate was synthesized by CobA. We used the same stop-flow method to detect the consumption of ATP in the CBAD-CobR-CobA and CBAD-CobR-CobA-CbiP reactions, whereby 0.05 mM and 0.15 mM ATP were calculated to be consumed by the CobA reaction and CobA-CbiP cascade reaction, respectively (Fig. 6e). In addition, we tried to strengthen the reaction module so as to be able to develop a directly detection method of AdoCBAD and AdoCby by LC-MS (Fig. 6f). The mass charge ratios (m/z) of AdoCBAD and AdoCby were detected with one positive charge were

1186.6263 and 1182.8359 using UPLC-MS, respectively. However, the corresponding species were practically non-detectable in negative ion mode. To further confirm the photosensitivity of the synthesized AdoCby, the sample was irradiated with 280 lux of light at 455 nm for 1 h and detected again by MS. As predicted, the MS signal of AdoCby disappeared due to the instability of the Co−C bond (Supplementary Fig. 7).

## Designing a cofactor regeneration module
Cell-free reaction system required additional recycling of cofactors and removal of by-products, since these functions are normally provided by a living cell but not an artificial in vitro system. The AdoCbl synthetic system developed in this study has a long and complex catalytic pathway with a wide variety of cofactors and by-products. Accordingly, a regeneration system was designed to replenish SAM, NADH, ATP, ʟ-glutamine, and 5-ALA, while decreasing the accumulation of by-products such as SAH, ʟ-glutamate, ADP, pyrophosphate and triphosphate in our cell-free reaction system (Fig. 7).

The glutamate dehydrogenase RocG from *Bacillus subtilis* was introduced to replenish the NADH demanded by CobR and HBA pathway, and also consume the accumulated ʟ-glutamate derived from the amidase reactions catalyzed by CobB and CbiP (Supplementary Fig. 8a). In addition to RocG, ʟ-glutamate was also catalyzed by glutamine synthetase GlnA from *E. coli* to replenish ʟ-glutamine as the amide donor needed by the synthetic pathway. The transformation yields of

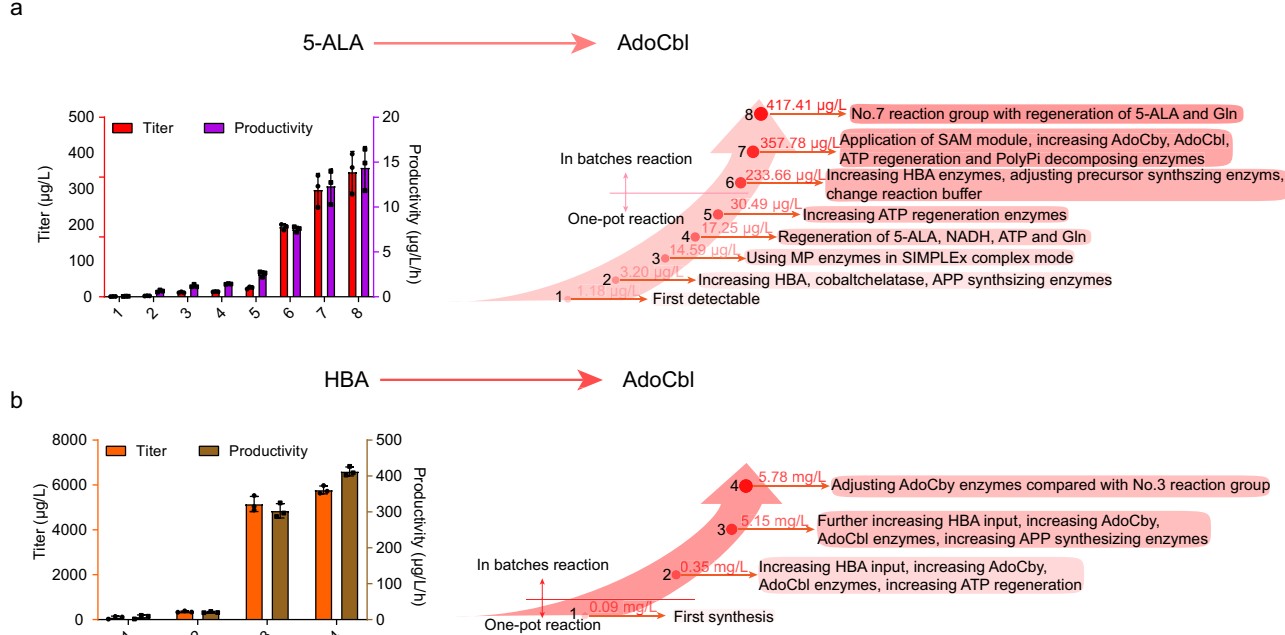

**Fig. 8 | Synthesis of AdoCbl in vitro. a** Titer and productivity of the cell-free reaction system from 5-ALA to AdoCbl during optimization. Reactions were performed in triplicate ($n = 3$ biologically independent samples) and data are presented as mean values ± SD. **b** Titer and productivity of the cell-free reaction system from purified HBA to AdoCbl during optimization. The significant strategy causing the improvement of every reaction system was highlight in the right axis of evolution, and specific reaction composition were listed in supplementary Table 2. Reactions were performed in triplicate ($n = 3$ biologically independent samples) and data are presented as mean values ± SD. Source data are provided as a Source Data file.

RocG and GlnA were measured in buffers with different pH values, and the consumption ratio of L-glutamate between RocG and GlnA was 2:1 at pH 8.0, representing the practical regeneration ratio of these two enzymes in simulative AdoCbl synthesizing system that formed of 2 mM L-glutamate, 1 mM NAD+ and 1 mM ammonium (Supplementary Fig. 8b).

Triphosphate was reported could exert strong feedback inhibition on CobA at more than concentration of 1 mM[41]. To eliminate inhibition by triphosphate, inorganic pyrophosphatase PpA from *E. coli* was introduced to decompose triphosphate into phosphate. We recorded the kinetics of *Ec*PpA in the reaction buffer, whereby the Km for triphosphate was 0.79 mM and the Vmax was 46.11 mM mg$^{-1}$ min$^{-1}$, indicating that *Ec*PpA could effectively prevent the accumulation of triphosphate to make sure the concentration of triphosphate under the obvious inhibition according to reported before[41] (Supplementary Fig. 8c). Because of the large consumption of ATP in the synthesis pathway and regeneration modules, we chose polyphosphate kinase PpK from *Corynebacterium glutamicum*[42] to regenerate ATP from ADP using externally added polyphosphate. Under the reaction conditions of the synthetic system, the Vmax for ATP synthesis was 4.46 mM mg$^{-1}$ min$^{-1}$ and the Km for sodium hexametaphosphate was 0.63 mM, working as the reference of input in our cell-free reaction system (Supplementary Fig. 8c).

To optimize the whole complex AdoCbl synthesis system and using the 2-oxoglutarate formed by RocG in the process of regenerating NADH, we extraly introduced the 2-oxoglutarate dehydrogenase complex OGDH from *E. coli* and 5-aminolevulinic acid synthase HemA from *R. sphaeroides* to design a complete NADH and 5-ALA regenerating module 1 to avoid the other byproduct accumulating in regenerating module (Fig. 7). The regeneration system was designed and applied throughout the whole in vitro system, which could regenerate the consumed substrate and cofactors partly and decreased the total standard Gibbs free energy change to −685.84 kcal/mol of synthetic pathway (Fig. 2). In the designed regeneration system, module 1 mainly regenerates NADH to make sure the reducing force supply of reaction system, and also recovered 5-ALA in case of accumulation of another by-product. Module 2 regenerates L-glutamine catalyzed by GlnA while consuming L-glutamate and ammonium, but requires a same stoichiometric amount of ATP. Module 3 is a key power engine to regenerate ATP from ADP by consuming inexpensive polyphosphate. Triphosphate and pyrophosphate accumulated in the reaction system were degraded by PpA in module 4 to relieve the corresponding inhibition. Finally, SAM module works to synthesize SAM from L-Met and remove SAH from reaction system by MetK and MtnN. These five regeneration modules only require the external input of glycine, polyphosphate and a small amount of CoA to start the reaction, but could partly regenerated 5-ALA, L-glutamine and NADH, while also preventing the accumulation of by-products to optimize the reaction system.

The regenerating modules were detected to enhance the cell-free system in synthesizing vitamin B$_{12}$ and intermediates by partly or completely introduced. In order to overcome the reported feedback inhibition of triphosphate and diphosphate on CobA enzyme, PpA enzyme was utilized in all reactions system outlined in Fig. 8 and Supplementary Table 2. The SAM module, comprising the MetK and MtnN enzymes, was able to increase the HBA titer by 3.3-fold in early experiments (Fig. 5a). Additionally, the combination of ATP, NADH and L-glutamine regeneration was found to improve the AdoCby module reaction (AdoCby synthesis using HBA as a substrate) by 3.0-fold (Supplementary Fig. 8d). The synthesis of AdoCby form HBA was also found to be promoted by only introducing ATP regeneration by 2.16-folds (Supplementary Fig. 8d), while in one-pot reaction producing AdoCbl from 5-ALA, increasing PpK enzymes input could promote the final titer of AdoCbl by 75% (No. 4 and No. 5 reaction groups in Fig. 8a). Furthermore, the introduction of the entire regeneration module resulted in a one-fold increase in the titer of AdoCbl (No. 3 and No. 5 reaction groups in Fig. 8a). In batch reactions (5-ALA to HBA, HBA to AdoCbl), the addition of regeneration module 1 increased the titer of AdoCbl from 357.78 to 417.41 µg/L (No. 7 and No. 8 reaction groups in Fig. 8a), representing a 16.7% increase.

**Promoting the reactions catalyzed by transmembrane enzymes**

In the AdoCbl synthesis pathway, adenosylcobinamide-phosphate synthase CbiB and AdoCbl-5′ phosphate synthase CobS involved in nucleotide loop assembly are irreplaceable integral membrane enzymes[43,44]. To improve their protein stability and overexpression, CbiB from *P. freudenreichii* and CobS from *E. coli* were co-expressed in the previously reported amphipathic protein complex SIMPLEx(-enzymes) form[45,46]. After fusion with the green fluorescence protein (GFP) as a reporter, fluorescence imaging showed that CbiB-GFP was expressed around the cell membrane, while SIMPLExCbiB-GFP was distributed in the whole cell and trended to gather at the cell poles (Supplementary Fig. 9a). The SDS-PAGE, western blot and GFP fluorescence analyses showed that these two transmembrane proteins had better expression in the form of SIMPLEx protein complexes (Supplementary Fig. 9b, c). The catalytic activity of SIMPLExCobS was tested in situ in an artificial plasmid-free AdoCbl synthesizing *E. coli* B16, which showed that the reconstructed strain with SIMPLExCobS successfully synthesized vitamin $B_{12}$, and the amphipathic protein complex even slightly increased the biomass as well as the product titer (Supplementary Fig. 9d).

**Synthesis of AdoCbl from 5-ALA**

The four common forms of cobalamin are AdoCbl, OHCbl, MeCbl, and CNCbl, of which AdoCbl is the active coenzyme form of cobalamin synthesized by biochemical catalysis in vivo, and CNCbl is the stable transformed type applied in industrial manufacturing, in which the adenosine group is replaced with a cyano group by treating with cyanide, significantly increasing the photostability compared to AdoCbl. The direct detection of cell-free synthetic AdoCbl was carried out using UPLC-MS and results confirmed that it had the same detecting signal as AdoCbl standards (Supplementary Fig. 10a). Typically, AdoCbl and its derivates are usually detected after converting adenosine into a cyano group using cyanide, as a means of stabilizing the compounds. However, to avoid the use of dangerous cyanide for the transformation of synthetic AdoCbl, it was discovered that a combination of sodium nitrite, acetic acid and crude cell extract of *E. coli* MG1655(DE3) in a boiling water bath could effectively convert AdoCbl into CNCbl without the need for cyanide (Supplementary Fig. 10b, e). The transformation method above is strictly dependent on the presence of bacterial cells, and the yield of transformation is directly related to the quantity of the strain used (Supplementary Fig. 10c, d, f–h). In addition, this transformation method was also expanded to include the cyanation of AdoCby, a vitamin $B_{12}$ derivative, to form the cyano-cobyrinate through the same substitution reaction between adenosyl group and cyano group (Supplementary Fig. S11). It is hypothesized that the cyano group synthesized from the reaction between phenol, polyphenol or analogue in bacterial cells and nitrite under acidic conditions[47] take the place of the adenosyl group in vitamin $B_{12}$ and its derivatives during transformation.

After assembling all of the synthetic and cofactor regeneration modules, we initiated the synthesis of AdoCbl in a cell-free reaction system using 5-ALA as a substrate. The initial detectable titer of AdoCbl was only 1.18 μg/L, with the productivity was 0.059 μg/L/h over a 12-h reaction process. However, by increasing the input of enzymes involved in the HBA pathway, cobalt-chelation, and utilizing the transmembrane enzymes in SIMPLEx complex mode, the titer increased to 3.20 and 14.59 μg/L, respectively. The introduction of regeneration module 1 through 3 led to a titer increase to 17.25 μg/L, and further strengthening the ATP regeneration by increasing PpK input promoted the titer to 30.50 μg/L. To achieve a significant improvement in titer, the in-batch reaction mode was utilized, synthesizing HBA from 5-ALA initially, and then adding the AdoCbl synthesizing enzymes to produce AdoCbl. Through optimization via the combination of in-batch reaction system and increasing HBA synthesizing enzymes, the titer of AdoCbl was

increased to 233.66 μg/L. Finally, with the introduction of the SAM module to the in-batch reaction system, along with an increase in enzymes input at various stages, the titer of AdoCbl from 5-ALA reached 357.78 and 417.41 μg/L, with and without regeneration module 1, respectively. These values were 303.20- and 353.74-fold higher than the corresponding values of the first reaction system, representing a productivity of 12.33 and 14.39 μg/L/h (Fig. 8a).

After discovering the strong promotional effect of using batch reaction technique, we aimed to further evaluate the flexible optimization of the cell-free AdoCbl synthesis system, to improve the AdoCbl titer by using HBA as a substrate, in order to simplify the complicate multiple-step cobalamin synthesis pathway. By adding input enzymes for potential bottleneck reactions and implementing an optimization strategy, we were able to increase the AdoCbl titer from 93.54 μg/L (an original synthetic reactant using HBA as substate) to 5.78 mg/L, while also increasing the productivity from 7.80 μg/L/h to 412.77 μg/L/h. The final yield was increased to 9.15% of the theoretical conversion yield, and all titer, productivity, reaction compositions were illustrated and listed in Fig. 8b and Supplementary Table 2. The synthesized AdoCbl was detected and identified using LC-MS and a bioassay based on *S. typhimurium* AR2680 (Supplementary Fig. 12)[48,49].

## Discussion

Due to its complicated structure and low yield from chemical synthesis, large scale industrial production of vitamin $B_{12}$ relies on microbial fermentation. However, both industrial strains and engineered strains have drawbacks either in the fermentation process or strain evolution, which has created a need for exploring and analyzing synthetic pathways, as well as innovative synthesis methods. By assembling every synthesis module, screening the bottleneck reactions, developing intermediate preparation and detection methods, and designing a customized cofactor regeneration system, we have introduced a cell-free reaction system as a promising alternative for AdoCbl production from both 5-ALA and HBA, achieved 0.42 mg/L and 5.78 mg/L of AdoCbl, respectively, which also presented the emergence of metabolic problem of assembling a synthesis pathway with multi-enzymes, multi-cofactors, and intricate relationships of mutual regulation in vitro.

In nature, vitamin $B_{12}$ is synthesized through an ancient and intricately regulated biosynthetic pathway, involving multiple cascade reaction with intricate interactions. Corresponding to that, while assembling and optimizing the cell-free production system for vitamin $B_{12}$, we encountered the emergence of metabolic problem that did not show up specifically in partial cascade reactions, but became pronounced after assembling all synthesis modules together. This issue hindered the promotion of final titer and yield, which conclude the imbalance of cascade reactions, competition of cofactors, feedback inhibition, as well as the instability of intermediate products. In addition, due to the lack of commercial sources and direct detection methods of intermediates, the screening of accumulated intermediates was also hardly to carried out. However, several strategies have been proposed to address these issues. For instance, after discovering the incomplete catalysis of CobB reaction, it was found that the strong driving force of CBAD reaction can shift the CobB reaction equilibrium towards complete reaction and push the metabolic flux towards downstream reaction (Supplementary Fig. 5c). In addition, when using HBA as substrate, the titer of AdoCby was observed to be higher in batches reaction mode compared to one-pot reaction mode, which was ascribed to the large demand for ATP in the AdoCby synthesis module, and highlighted the necessity of cofactors regeneration or multi-pot strategy (Supplementary Fig. 8d). Furthermore, the reaction pathway has been reported to be subjected to feedback inhibition on multiple levels, such as Urogen III inhibits the activity of uroporphyrinogen III $^{(C2,7)}$-methyltransferase CobA[6], HBAD and

triphosphate can inhibit adenosyltransferase CobA[50], while the by-product S-adenosyl-homocysteine inhibits most methylases[51]. To overcome these inhibitions, we introduced additional enzymes for the regeneration or removal of accumulated inhibitors, making reaction system viable. In addition, some intermediate products of this pathway are unstable and easily oxidized, such as the precorrin-derived precursor of HBA[6] and intermediate from CBAD to AdoCby in the synthesis module, as a result, it is necessary to use a reductant or an anaerobic glove box in specific reactions for their production and analysis. Furthermore, during the later stages of further promoting the final titer, it is possible that the pH of the buffer environment may change as a result of the prolonged reaction and the accumulation of hard-removed byproducts such as $CO_2$, altering the activity of enzymes (Supplementary Fig. 13). As a result, it is crucial to prioritize the screening of optimal reaction buffers or to consistently control the pH to achieve further optimization or amplification[52].

Nowadays, the production of vitamin $B_{12}$ on a commercial scale relies heavily on bacteria fermentation utilizing *P. denitrificans*, *P. shermanii* or *S. meliloti*[53,54] or places hope on developing engineered type strain such as *E. coli*[10]. While mature fermentation protocols offer superior vitamin $B_{12}$ titers of more than 200 mg/L, the available industrial strains require lengthy fermentation periods (more than 8 days), and lack efficient genetic engineering tools (limited in random mutagenesis and plasmid-based gene expression) to further increase production[54]. Even the introduction of the CRISPR-based genetic tool to *S. meliloti*, which broke the genetic modification obstacle of industrial strains, only resulted in a modest 10%-25% promotion and approximately 100 mg/L titer due to the complex relationship between cell growth and vitamin $B_{12}$ production[55]. Although engineered *E. coli* has reduced the fermentation period and simplified genetic modification compared with industrial strains, reported vitamin $B_{12}$ titer based on *E. coli* was less than 1 mg/L due to largely introduced heterogenous genes and complex, elaborately regulated synthetic pathway. Thus, in terms of titer and productivity, cell-free synthesis of vitamin $B_{12}$ has the potential to compete with bacteria fermentation. Although the average productivity of industrial fermentation is calculated as 1.04 mg/L/h (final titer as 200 mg/L and fermentation period as 8 days), achieved productivity in this manuscript of cell-free synthesis was 0.41 mg/L/h (final titer as 5.8 mg/L and reaction period as 14 h), which is comparable to bacteria fermentation. In addition, cell-free system can bypass difficult and time-consuming genetic engineering and directly operate enzymes, cofactors and buffers, shortening the optimization period. However, in the terms of input-out ratio, cell-free synthesis system is currently not advantageous compared with bacteria fermentation due to the high cost of protein purification and cofactors consumption. Nevertheless, enhancing enzyme activity by immobilization[56–58] or innovating in the co-purification process[59,60] could significantly reduce the cost of protein purification, making cell-free synthesis system more competitive with bacteria fermentation in terms of input-output ratio and industrial application in the future.

Although the titer of vitamin $B_{12}$ needs to be further improved to make the process industrially feasible, the strategies of developing the cell-free 36-enzyme reaction system described here will be useful for constructing other cell-free enzyme reaction systems for the in vitro production of complex molecules.

## Methods

### Cloning and purification of enzymes

The enzymes of the HBA pathway, including CobG from *B. melitensis* bv.1 str. 16 M (gift from Q. M. Wu, China Agricultural University), as well as CobA, CobI, CobK, CobJ, CobM, CobF, CobK, CobL, and CobH from *R. capsulatus* SB 1003 (gift form D. K. Newman, California Institute of Technology), were heterologously co-expressed in a pET28a-derived plasmid controlled by the lac operon and T7 promoter, or separately expressed in a pET28a-derived plasmids. The *cobS* and *cobT* genes from *S. meliloti* 320[5], which encode the subunits of cobaltochelatase, were co-cloned into the pACYCDuet-1 vector and co-expressed in a single strain to ensure their stability and activity. The membrane proteins CobS from *E. coli* MG1655 and CbiB from *P. freudenreichii* ATCC 9614 (gift from Y. F. Zhang, Tianjin Institute of Industrial Biotechnology, CAS) were cloned into the soluble complex expression vector by fused with ApoAI* and ΔSPMBP sequence (Synthetic gene from GENEWIZ, China) as reported before[45,46]. The coding gene of CbiA from *M. jannaschii* was synthesized by GENEWIZ (China) according to CBIA_METJA (https://www.uniprot.org/uniprotkb/Q58816/entry), the peptide sequence from Uniprot database (https://www.uniprot.org/). Other enzymes were cloned from corresponding strains described in Supplementary Table 1 into pET28a, pACYCDuet-1 or pCDFDuet-1, in which *R. sphaeroides* 2.4.1 is a gift from H. Liu (Tianjin University of Science and Technology), *B. megaterium* CICC10055 is a gift from J. Liu (Tianjin Institute of Industrial Biotechnology, CAS), *P. denitrificans* ATCC 13867 was purchased from Xiangf Bio (China), *C. glutamicum* ATCC 13032 is a gift from C. H. Bi (Tianjin Institute of Industrial Biotechnology, CAS), *S. typhimurium* AR2680 is a gift from M. J. Warren (University of Kent), and *E.coli* MG1655, *B. subtilis* 168 are laboratory storage strains. Unless specified otherwise, all enzyme-encoding plasmids were introduced into *E. coli* BL21(DE3), and enzymes were expressed in LB medium with 50 μg/ml kanamycin, 50 μg/ml streptomycin or 34 μg/ml chloramphenicol. The 600 ml cultures were inoculated with 6 ml of a seed culture cultivated overnight in the same medium, and grown to an $OD_{600}$ of 0.6–0.8 at 37 °C. The cultures were then induced with 0.8 mM isopropyl-β-d-thiogalactoside (IPTG), and allowed to express the recombinant enzymes at 16 °C for 20 h. The cells were harvested by centrifugation at 1750*g*, and resuspended in 20 ml binding buffer (20 mM $NaH_2PO_4$, 500 mM NaCl, and 30 mM imidazole, pH 7.4). The cells were lysed using a high-pressure cell homogenizer. The unbroken cells and cell debris were removed by centrifugation at 10,960 *g* for 1 h. The cleared lysate was incubated with Ni-NTA resin for 1 h at 4 °C. The resin was washed with 5 column volumes of wash buffer (20 mM $NaH_2PO_4$, 500 mM NaCl, and 100 mM imidazole, pH 7.4) three times. The protein was eluted with 5 ml of elution buffer (20 mM $NaH_2PO_4$, 500 mM NaCl, and 500 mM imidazole, pH 7.4). The eluted enzymes were buffer-exchanged into protein storage buffer (50 mM Tris-HCl, pH 8.0) by ultrafiltration, and 20% glycerin was added as a cryoprotectant. The aliquots were stored at −80 °C, and the glycerin was removed before use by ultrafiltration. The enzyme concentrations were determined using the Bradford method.

### Purification of key intermediate products

Uroporphyrinogen III was synthesized by cascade enzyme catalysis based on a previous report[23]. A 5 ml reaction mixture containing 0.32 μM HemB, 2.27 μM HemC, 2.12 μM HemD, 5 mM ALA and 200 μM SAM was incubated in an anaerobic bottle sparged with $N_2$. After reaction for 2 h at 30 °C, 15 ml of methanol was added to terminate the reaction and the precipitated protein was removed by centrifugation. HBA and HBAD was purified by Ni-NTA resin using the enzyme-trap approach[6] and released from binding protein by boiling water bath for 30 min. The Uroporphyrinogen III was placed in a boiling water bath for 30 min to induce complete oxidation, and quantified using $ε_{405} = 548$ mM$^{-1}$ cm$^{-1}$[24]. HBA and HBAD were quantified using $ε_{329} = 49.6$ mM$^{-1}$ cm$^{-1}$ and $ε_{323} = 50$ mM$^{-1}$ cm$^{-1}$, respectively.

### HBA and precursor module reaction system

The original HBA module reaction system contained 0.1 μM HemB, 1 μM HemC, 1 μM HemD, 10 μM CobA, 5 mM ALA, 1 mM SAM, 2 mM NADH, 1 mM NADPH, 5 mM $MgCl_2$, 10 mM KCl, 5 mM NaCl, HBA crude cell extract of 155.8 mg/ml wet cell weight HBA cell culture fluid (corresponding 17.81 mg/ml dry cell weight, 6.23 mg/ml total protein, or 10 $OD_{600}$ HBA cell culture fluid per volume reactant, and the detail

information of HBA crude cell extract was illustrated in Supplementary Fig. 11). The lysate was washed twice with 50 mM Tris-HCl (pH 8.0) using a 10 kDa MWCO Millipore ultrafiltration cartridge, and adjusted to an appropriate volume as a crude cell mixture. SAM and 5-ALA were added 1 mM every hour in fed-batch mode over 5 h, or added 1 mM and 5 mM once in the 12 h reaction system.

The optimized HBA module reaction system contained 0.1 μM HemB, 1 μM HemC, 1 μM HemD, 10 μM CobA, 5 μM MetK, 5 μM PpK, 10 μM MtnN, 2 mM AMP, 1 mM SMPP, 5 mM ALA, 1 mM L-Met, 0.236 mM NADH, 0.236 mM NADPH, 5 mM MgCl₂, 10 mM KCl, 5 mM NaCl, HBA crude cell extract of 155.8 mg/ml wet cell weight HBA cell culture fluid per volume reactant (detail information was illustrated in Supplementary Fig. 11) The crude cell extract was treated as described before.

### Enzyme activity assay of hydrogenobyrinate *a, c*-diamide synthase

The enzyme activity assay was modified based on a previous report[32]. A reaction mixture containing 5 μM HBA, 1 mM ATP, 2.5 mM MgCl₂, and 1 mM L-glutamine in a total volume of 250 μl in 100 mM Tris-HCl (pH 8.0) was incubated at 32 °C for 60 min. The reaction was terminated by boiling water bath for 30 min and detected by HPLC.

### Enzyme activity assay of the cascade reaction between CobB and CobNST

A reaction mixture contained 5 μM HBA, 3 μM CobB, 5 μM CobN, 5 μM CobST, 2 mM L-glutamine, 5 mM MgCl₂, 10 mM KCl and 5 mM NaCl in 100 mM MOPS buffer (pH 8.0), to which we added 0.1 mM CoCl₂ and titration of ATP at 1 mM, 5 mM, 10 mM or 0.5 mM ATP, 1 mM AMP, 5 mM SMPP and 3 μM PpK; 5 mM ATP and 0.05 mM, 0.1 mM, 0.2 mM, 0.5 mM CoCl₂. The reaction was terminated by boiling water bath for 30 min and detected by HPLC.

### Enzyme activity assay of CobR

A reaction mixture containing 0.5 mM FMN, 5 mM NADH and 5 μM CobR or 5 μM Fre or 5 μM CobR, 5 μM CobA or 5 μM CobR, 5 μM CobA, 5 μM Fre in Tris-HCl (pH 8.0) was used to detect the FMN-reduction activity of CobR and enzyme combinations. FMN was monitored by the absorbance at 450 nm and the conversion was calculated by comparing with a standard curve.

Reaction mixtures containing 5 mM NADH, 5 μM CobR and different concentrations of FMN in 100 mM Tris-HCl (pH 8.0) were used to record the enzyme kinetic parameters. FMN was monitored in intervals of 5 s in each concentration to calculate the catalytic velocity, and the data were fitted to the Michaelis-Menten equation.

To detect the CBAD reducing reaction catalyzed by CobR, a mixture containing 0.5 mM FMN, 5 mM NADH and 5 μM CobR (or 5 μM CobR, 5 μM CobA and 1 mM ATP) in Tris-HCl (pH 8.0) was incubated for 3 min. In addition, CBAD-control (CBAD reaction system without the addition of HBAD) or CBAD were introduced and the recovery of absorbance at 450 nm was monitored to calculate the change of FMN concentration.

### Stop-flow reaction method in the AdoCby module

The CobR-CobA reaction mixture contained the CBAD reactant (final concentration of HBAD was 1 μM), 1 μM CobR, 1 μM Fre, 1 μM CobA, 0.5 mM FMN, 5 mM NADH, and 1 mM ATP in Tris-HCl (pH 8.0). The CobR-CobA-CbiP reaction mixture contained CBAD reactant (final concentration of HBAD was 1 μM), 1 μM CobR, 1 μM Fre, 1 μM CobA, 1 μM CbiP, 0.5 mM FMN, 5 mM NADH, 1 mM ATP and 1 mM L-glutamine in Tris-HCl (pH 8.0). The reaction mixtures were incubated at 32 °C for 5 h. Then, 1 μM PpA was added to hydrolyze triphosphate into phosphate. Samples were diluted to an appropriate concentration, after which ATP was detected using an ATP Assay Kit and phosphate was detected according to a previous report[61].

### Synthetic reaction system of AdoCBAD and AdoCby

The reaction mixture containing 10 μM HBA, 3 μM CobB, 15 μM CobN, 10 μM CobST, 3 μM PpK, 2 mM L-glutamine, 10 mM AMP, 10 mM SMPP, 0.2 mM CoCl₂, 5 mM MgCl₂, 10 mM KCl and 5 mM NaCl in 100 mM MOPS buffer (pH 8.0) was incubated for 2 h at 32 °C to generate the CBAD reactant. Then, 90 μM CobR, 9 μM Fre, 90 μM PpA, 30 μM CobA, 2 mM NADPH, 5 mM NADH, 0.1 mM FMN, 2 mM L-glutamine, 5 mM ATP, 5 mM DTT and the CBAD reactant were incubated in the anaerobic chamber for 12 h at 32 °C in the dark to synthesize AdoCBAD. In addition, 90 μM CbiP was added to synthesize AdoCby under the same conditions.

### Analytical methods

Uroporphyrin III was detected using an Agilent 1200 HPLC equipped with a ZORBAX SB-Aq column (4.6 × 150 mm 5 μM, Agilent) and interfaced with a Bruker microQ-TOF II mass spectrometer. The uroporphyrin III standard was quantified using $\varepsilon_{405} = 489\ mM^{-1}\ cm^{-1}$ and samples were analyzed by measuring the absorbance at 405 nm. Uroporphyrinogen III was measured after it was completely oxidized to uroporphyrin III by incubation at 100 °C for 30 min.

Precorrin-2 was converted into sirohydrochlorin by incubating with 1 μM SirC and 1 mM NAD⁺ at 30 °C for 20 min, and measured using fluorescence (excitation wavelength: 378 nm, emission wavelength: 600 nm).

HBA and HBAD were measured using HPLC as reported previously[10], or using a Waters ACQUITY UPLC H-Class system equipped with a PDA diode array detector, QDA mass spectrometry detector and an XBridge BEH C18 column (3.0 × 100 mm 2.5 μM, Waters). Samples were analyzed at 35 °C and monitored at 329 nm. The mobile phase was composed of water containing 0.1% formic acid (solvent A) and methanol containing 0.1% formic acid (solvent B) at a flow rate of 0.5 ml/min. The elution gradient was as follows: 10–30% B (0–3 min), 30%B (3–5 min), 30–100% B (5–7 min), 100% B (7–7.5 min), 100–10% B (7.5–8 min), 10% B (8–10 min). All samples for chromatography were passed through a 0.22 μm pore-size filter.

ATP was diluted to an appropriate concentration (0.1–10 μM) and measured using an ATP assay Kit (S0026, Beyotime biotechnology) according to the manufacturer's protocol.

AdoCby and AdoCBAD were produced in relevant reaction systems, and the proteins were precipitated by adding a three-fold volume of ethanol. Samples were centrifuged at 13,800 g to remove the precipitated protein, and the supernatant was dried under vacuum at 35 °C. The dried samples were resuspend using the same volume of ddH₂O and passed through a 0.22 μm pore-size filter before UPLC-MS analysis. All steps were performed in the dark. UPLC-MS detection of AdoCBAD and AdoCby was performed using a Waters ACQUITY UPLC H-Class system equipped with a PDA diode array detector, QDA mass spectrometry detector and a XBridge BEH C18 column (3.0 × 100 mm 2.5 μM, Waters). Samples were analyzed at 35 °C. The mobile phase was composed of water containing 0.1% formic acid (solvent A) and methanol containing 0.1% formic acid (solvent B) at a flow rate of 0.3 ml/min. The elution gradients were: 10–30% B (0–3.5 min), 30%B (3.5–6.5 min), 30–100% B (6.5–8.0 min), 100% B (8.0–9.5 min), 100–10% B (9.5–12 min), 10% B (12–15 min).

Triphosphate was converted into phosphate using 5 μM PpA before measurement as reported before[61]. FMN and NADH were quantified by measuring the absorbance at 450 and 340 nm, respectively.

The CNCbl standard was directly injected into an AB Sciex 5600 Triple TOF mass spectrometry detector. AdoCbl was transformed to the cyano-form by treatment with 10% v/v sodium cyanide (1% w/v) irradiated with 270 lux of light at 455 nm for 30 min or treatment with 10% v/v sodium nitrite (8% w/v), 10% v/v acetic acid and an appropriate amount (OD₆₀₀ = 20) of wild type *E. coli* MG1655 (DE3) wet cells at boiling water bath for 30 min, and measured using an Agilent 1200

HPLC equipped with a ZORBAX SB-Aq column (4.6 × 150 mm 5 μM, Agilent) and interfaced with a Bruker microQ-TOF II mass spectrometer or a Waters ACQUITY UPLC H-Class system equipped with a PDA diode array detector, QDA mass spectrometry detector and a XBridge BEH C18 column (3.0 × 100 mm 2.5 μM). Samples were analyzed at 35 °C and monitored at 361 nm. The mobile phase was composed of water containing 0.1% formic acid (solvent A) and methanol containing 0.1% formic acid (solvent B) at a flow rate of 0.5 ml/min. The elution gradient was as follows: 30% B (0–5 min), 30–100% B (5–6 min), 100% B (6–7 min), 100–30% B (7–7.5 min), 30% B (7.5–10 min). For separating the different forms of cobalamin, the elution gradient was also changed as follows: 10–30% B (0–5 min), 30–100% B (5–6 min), 100% B (6–7 min), 100–30% B (7–7.5 min), 30% B (7.5–10 min). All chromatography samples were passed through 0.22 μm pore-size filters. AdoCbl was also detected using a bioassay based on *S. typhimurium* AR2680 (gift from M. J. Warren, University of Kent) as reported previously[48,49]. Cells from a 10 ml overnight culture of *S. typhimurium* AR2680 in LB medium were collected by centrifugation, washed twice with sterile 0.9% v/v NaCl solution and resuspended in 5 ml of the same solution. This cell suspension was mixed with 200 ml minimal agar medium (0.5 g/L NaCl, 6 g/L $Na_2HPO_4$, 3 g/L $KH_2PO_4$, 1 g/L $NH_4Cl$, 4 g/L glucose, 2 mM $MgSO_4$, 0.1 mM $CaCl_2$, 50 mg/L cysteine) with 50 μg/ml streptomycin, and wells were made in each agar plate. Then, 20 μl samples were loaded into the wells and incubated overnight at 37 °C kept in dark.

All plotted data in the figure and Supplementary figure was analyzed using GraphPad Prism (version 9) or OriginPro (version 9.1). The chemical structure was drawn using ChemDraw (version 20.0). Multi-panel figures were arranged using Adobe Illustrator 2022.

### Reporting summary

Further information on research design is available in the Nature Portfolio Reporting Summary linked to this article.

## Data availability

A reporting summary for this article is available as Supplementary Information file. The main data supporting the findings of this study are available within the article and its Supplementary figures. The source data underlying Figs. 2–6, Fig. 8, Supplementary Figs. 1–3, Supplementary Figs. 5–14 are provided as a Source Data file. Specific data *P*-values are also included within the Source Data file. Additional details on datasets and protocols that support the findings of this study will be made available by the corresponding author upon request. Source data are provided with this paper. All the databases used in the study: Uniprot database: https://www.uniprot.org/, MetaCyc: https://metacyc.org/, eQuilibrator: https://equilibrator.weizmann.ac.il/, AlphaFold Protein Structure Database: https://alphafold.ebi.ac.uk/. Source data are provided with this paper.

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

## Acknowledgements

S.Y.L. is the chairman of the scientific advisory board of the Tianjin Institute of Industrial Biotechnology. We also thank Y.H.Z. and Z.Z. for insightful discussions and suggestions. D.Z. was supported by the National Key R&D Program of China (2020YFA0907800), the National Natural Science Foundation of China (22178372), National Science Fund for Distinguished Young Scholars (22325807), Science and Technology Partnership Program, Ministry of Science and Technology of China (KY202001017) and the Tianjin Synthetic Biotechnology Innovation Capacity Improvement Project (TSBICIP-KJGG-011). H.F. was supported by the Youth Innovation Promotion Association, Chinese Academy of Sciences (2020182). S.Y.L. was supported by the Tianjin Synthetic Biotechnology Innovation Capacity Improvement Project (SBICIP-CXRC-055), and the Development of platform technologies of microbial cell factories for the next-generation biorefineries project (2022M3J5A1056117) by the Korean Ministry of Science and ICT through the National Research Foundation.

## Author contributions

Q.K., H.F., SY.L., and D.Z. conceived and designed the experiments; Q.K., M.X., K.X. performed the experiments. H.F. and P.J. helped guide the pathway analyzing and gene editing; Q.K., C.Y., and D.Z. analyzed the experimental data; Q.K., SY.L., and D.Z. wrote the manuscript. All authors approved the final manuscript.

## Competing interests

The authors declare no competing interests.
