## [Peer Review File · Nature Communications]

A synthetic cell-free 36-enzyme reaction system for vitamin B12 productionREVIEWER COMMENTS

Reviewer #1 (Remarks to the Author):

The manuscript presented by King et al. presents a cell-free reaction system for the production of adenosylcobalamin. This 36 enzyme pathway was split into 5 modules which the authors individually tuned before combining for synthesis from 5-aminolevulinic acid. I am impressed with the work presented in this manuscript and believe that it will be significant for the synthetic biology and biochemistry communities. While individual components of this pathway have been explored in cell-free systems, the assembly of this complete pathway is novel and noteworthy. I believe that the experimental design is sound and only have a few comments for clarification that I think need to be addressed in the manuscript.

- It isn't clear what was done purified and what was done in crude lysates. The authors should specify throughout the manuscript which enzymes are purified and which enzymes are crude. Is Fig. 4 the only crude lysate? Is the final system a mixture?

- line 182 the reference to Fig 1a should be fig 3a?

- Fig. 4, What is the final concentration crude lysate? 10 g of HBA extract should be written as a concentration. Also, is that total protein or heterologously expressed protein?

- line 264, the authors decide to modify buffer/pH but the rationale for testing this isn't clear. What was the authors hypothesis? Is there previous literature to suggest that this would be an important lever in the cell-free environment?

- Fig. 5, What pH is being referred to in this figure? Is this initial pH of the reaction, final pH, buffer pH before assembly of the enzymatic reaction? If this is of the buffer, how does this impact the reaction? Please clarify.

- Fig. 6 is really small and difficult to read. The authors should consider reformatting to increase figure size. Maybe move d to be first panel because it is the first part referenced followed by current panel a on the first row above b and c, that would make a 3 row and 2 column figure allowing for all panels to be bigger and readable.

- Fig. 8, this is a great figure. However, I don't understand the term strengthen. The authors should be explicit about what is changing or the difference at each stage. I appreciate the conditions listed in table 2, but the term strengthen is used several times and has different meanings in each case. Please use more specific words for what is intended.

Reviewer #2 (Remarks to the Author):

Overall Comments:

This paper describes an *in vitro* reconstitution of vitamin B12 biosynthesis from the early precursor 5-ALA, which is a complex process involving >30 enzymatic reactions. The authors document their optimisation of the reaction conditions, as well as describing how they have overcome challenges in supplying reactants and avoiding feedback inhibition by incorporating a 'regeneration module' (to regenerate key reactants and consume potent inhibitors) to increase the yield. The strategies employed here will be generally useful for the development of other cell-free synthetic pathways. It remains unclear whether such a cell-free system is ever likely to compete with bacterial fermentation for production of vitamin B12 on a commercial scale.

The authors highlight several challenges associated with metabolic engineering to improve B12 synthesis (lines 54-56 and 560-564). However, these challenges seem minor relative to those which remain if vitamin B12 is to be produced on a commercial-scale by cell-free synthesis, with a potential for the text to mislead readers:

For example, line 561 states (in reference to engineered *E. coli* strains for B12 production) 'titers, yield and productivity remain discouragingly low, and further optimization is a challenge', but this seems at odds with the authors apparent optimism regarding cell-free synthesis since the yields per litre are no better. The latter also seems to be a technically more challenging (expensive) manufacturing route.

The authors also describe an *E. coli* strain which produces 0.67 mg/L from 24 hrs fermentation, which seems promising in comparison to the 0.4 mg/L of B12 reported here using cell-free synthesis from an early precursor 5-ALA. Moreover, other engineered strains of *E. coli* have already generated a promising 2.9 mg/L (88 mg from 30 L culture) of an advanced intermediate HBAD, (Widner et al 2016, <https://doi.org/10.1002/anie.201603738>).

Specific Comments:

Line 54: Prior work of Warren et al in engineering B12 pathways in *E. coli* should also be cited (eg Widner et al 2016, <https://doi.org/10.1002/anie.201603738>).

Figure 1 is a very useful summary of the complete B12 synthesis process but the chemical structures are very small and chemical reactions very difficult to make out.

In Figure 1, it may be useful to label the branch modules 'Branch Module 1' (as it is) and 'Branch Module 2' (for reaction of 5,6-dimethylbenzimidazole to alpha-ribazole-5'-P) as it is not clear from first glance that these are two separate processes.

Line 250 says 'no obvious increase in the precorrin-2 titer was observed after additional SAM supplementation'. This does not make sense since Fig S2b shows an increase in precorrin-2 until > 5 mM SAM is added. Please clarify/revise.

Line 297: 'but CbiA were main catalyzed the HBA derivant cobyrrinate to cobyrrinate a,c-diamide in the natural anaerobiotic metabolic pathway'. The meaning of this sentence is unclear, please revise.

Figure 4c: Suggest changing the x-axis legend to 'SAM or L-Met' (currently it looks like it refers to the ratio of SAM/L-Met).

Lines 426 and 429: 'Stoichiometrically limiting reagent' is more precise than 'stoichiometric limiting condition'

Line 463: What is meant by 'stoichiometrically reduced the material waste'? Does this mean that there was no material waste? Please clarify.

Line 475: What is meant by 'stoichiometrically reduce the demand for 5-ALA...'? Please clarify.

Line 482: Correct 'extraly introducing' with 'the additional introduction of'

In Figure 8 'Productivity' is misspelt.

Line 518: There is insufficient data in Fig S9c to claim a 'linear dependence' on the quantity of strain. It would be acceptable to say 'the transformation yield is related to the quantity of strain'.

Line 576: 'the cascade reaction was titrated, raising the positive driving force but the challenge of competitive relation in cascade reactions'. Meaning is unclear, please revise.

The discussion section contains a summary of the work performed to optimise in vitro B12 synthesis but references to relevant data/figures are missing.

Supplementary Table 1:

It should be made clear how delta G has been calculated for each transformation (ie at pH 8, ionic strength 0.1M or at pH 7.3, ionic strength 0.25M?) as well as which explicit chemical reactions this delta G corresponds to.

Footnotes are denoted 'a' and '1' – please make numbering consistent.

In Figure S2 it is not obvious how conditions differ between panels b-d. It would be helpful for key differences in conditions to be evident on the panels themselves (by including a '+MtnN' label on panel d, for instance).

Figure S4 is unclear.

-What does 'consumption' refer to in each panel? Ie in panel 'a' is this consumption of HBA? ATP?

-There are two y-axes on each panel but it is not clear which data corresponds to which y-axis.

-In Fig S4c what do triangles represent? It is not clear what the x-axis represents? What does 'proportion' refer to? Please clarify.

Figure S8d: The right hand y-axis 'Titer/Biomass' – should the units be mg OD-1 (rather than mg-1 OD-1)?

Figure S10b: It is very difficult to see the bacterial growth in the images of Salmonella bioassay plates. Is a better resolution picture available?

Reviewer #3 (Remarks to the Author):

In the presented article “A synthetic cell-free 36-enzyme reaction system for vitamin B12 production” by Kang et. al., the authors describe an extensive and monumental effort to develop cell-free B12 vitamin production pathway and address major barriers for high titer/high productivity levels in vitro. This will be a significant addition to the field of synthetic biology and in vitro production of commodity chemicals as it demonstrates viability of cell free synthesis of exceptionally complex compound with input of diverse cofactors and efforts to recycle and regenerate these cofactors. While there was major work done to address the bottlenecks, to drive the flux of pathway downstream, the possible inhibition mechanisms and ways to address these issues, there are few revisions needed before this article is ready to be published. Overall, the work is substantial and adds to the toolkit of expanding field of cell-free biology.

1)HBA Module: why this module was processed via crude extract instead of purification of each enzyme? While it may be better to co-express all enzymes and utilize crude extract to synthesize products, the choice of lysate rather than pure enzymes seems to be a major pitfall of the project. The actual yield of HBA (and subsequent AdoCbl) is around 10% indicating that this module is the major bottleneck in B12 synthesis. The authors tried to address this low yield/titer by looking into feedback inhibition and reaction buffering. While SAM/SAH ratio plays a significant role in methyltransferase activity inhibition and authors cleverly overcome this obstacle with MetK/MtnN, the effect seems to be abolished past 3 mM SAM addition (Fig. S2). It seems that last 3 steps (CobK/CobL/CobH) are thermodynamically unfavorable in this module and the final equilibrium may be limiting the titer/yield. Authors need to address this possibility, maybe create a sink for HBA with introduction of RcCobB and see how the flux changes in this module.

2)These pot reactions are expected to be acidic over time (based on by-products such as CO₂) and authors changed the buffer to HEPES-NaOH to address this matter. While it may not be obvious initially, these incremental drops in pH destabilize the system and inactivate enzymes thereby lowering the titer and yields. Recent work in NC on cell-free isobutanol production (<https://doi.org/10.1038/s41467-020-18124-1>) showed that control of optimal pH in reactions considerably increased the titers and productivity of overall system. Authors need to clarify and address this obvious barrier that may significantly alter their overall system. In the future, they may also test better buffer systems (such as bicarbonate solutions).

3)Line 339 states that excessive ATP negatively affected the product titers which is surprising. Is there any particular reason for this observation? The ATP regeneration system seems to have no such adverse effect, probably due to titrating and optimizing PpK levels. If it is so, authors need to state these findings.

4)While regeneration system overall is elegant and increases the titers of overall system by ~16%, it is not stoichiometrically balanced and it seems to have more supplemental role rather than fully recycling/regenerating system. Moreover, the regeneration system may be compromised and needs to

be optimized further with the use of crude extract as lysates contain residual non-specific ATPase and NADH/NADPH oxidase activities that may alter the balance in the reaction. It is not clear whether increase in titer is due to overall module or parts of the system such as PpA and MetK/MtnN.

5) 5.78 mg/L were obtained after full assembly of the system but it is not clear if the pathway is truly one pot or the final product is synthesized via modules, intermediates are extracted to be further processed (such as AdoCBAD and AdoCby synthesis in anaerobic conditions). The volume of each reaction is not clear from methods sections and whether these findings would scale on a higher volume. It would be beneficial to have a table of final concentration of each enzyme used in the final assembly.

Minor Revisions:

Line 40: change to 5' deoxyadenosine

Line 213-214: CysG is at the end of operon and it is not clear what CysGA and CysGB refer to. That would help to explain fig. 4A better.

Line 293-305: while the authors performed genome mining for AdoCby module, they chose *S. meliloti* HemBCD genes in precursor module. Is there any reason/discussion for this approach? Given the immense size of the project, was genome mining done in ad hoc fashion?

Line 482-482: Needs grammatical revision.

Line 531-532: Fig. 8 labels: change the label to "productivity"

Line 966-967. Needs grammatical revision

Reviewer #4 (Remarks to the Author):

Kang and collaborators assembled a pathway for in vitro biosynthesis of vitamin B12 (adenosylcobalamin, AdoCbl). Authors performed a cascade of catalytic reactions that start from 5-aminolevulinic acid (5-ALA) as an inexpensive substrate. More than 32 biocatalytic reactions were integrated and optimized to achieve complete cell-free synthesis of AdoCbl, after overcoming feed-back inhibition, analytical issues, instability of intermediates, and imbalance and competition of cofactors. Authors report that this cell-free system produced 417.41 $\mu\text{g/L}$ and 5.78 mg/L of AdoCbl using 5-ALA and the purified intermediate product hydrogenobyrrinate as substrates, respectively. This study is interesting from a technical point of view (36 enzymes catalyzing 30/32 reactions), but neither the product nor the strategy are novel/interesting. All the enzymes have been extensively described in literature. All the biochemical steps have been thoroughly characterized by other researchers. Mixing

the enzymes together and measuring product formation does not come across as particularly disruptive. The regeneration system used by the authors is not novel, either, and has been used multiple times to reconstitute cofactors in cell-free systems (in example, ATP). Worryingly, the authors do not even know all the chemical transformations that happen in their system (Lines 514-518). In summary, while the technical feat of assembling a complex pathway has some interest, the level of novelty and impact in the biocatalysis community is limited. A more specialized journal (ACS Catalysis, Metabolic Engineering) is recommended as a venue for this study. Other comments:

+ Urogen should be synthesized under aerobic conditions. The cascade catalysis mentioned by the authors is not clear for this Reviewer. What is the stability of uroporphyrinogen? This should be reported.

+ L211: Large (???) plasmid?

+ All figures in the main text lack statistical comparisons, making the optimization of steps claimed by the authors dubious.

+ The mass relationships for all the enzymes used in the cascade is not explained. How did authors optimize this parameter (how much enzyme is added for each component)?

Dear Reviewers:

Thank you very much for your time and effort of reviewing our manuscript and providing valuable comments on our manuscript titled ‘*A synthetic cell-free 36-enzyme reaction system for vitamin B₁₂ production*’ (NCOMMS-23-03535-T). We have thoroughly revised our manuscript by addressing the points raised by the four reviewers.

Here, we resubmitted our revised manuscript (both clean version and change-tracked version) and response letter to all Reviewers. We hope that our revised manuscript is now suitable for publication in *Nature Communications*.

Reply to Reviewers’ comments point-by-point:

Reviewer #1 (Remarks to the Author):

The manuscript presented by Kang et al. presents a cell-free reaction system for the production of adenosylcobalamin. This 36-enzyme pathway was split into 5 modules which the authors individually tuned before combining for synthesis from 5-aminolevulinic acid. I am impressed with the work presented in this manuscript and believe that it will be significant for the synthetic biology and biochemistry communities. While individual components of this pathway have been explored in cell-free systems, the assembly of this complete pathway is novel and noteworthy. I believe that the experimental design is sound and only have a few comments for clarification that I think need to be addressed in the manuscript.

- It isn't clear what was done purified and what was done in crude lysates. The authors should specify throughout the manuscript which enzymes are purified and which enzymes are crude. Is Fig. 4 the only crude lysate? Is the final system a mixture?

- The Cob enzymes in the HBA synthetic pathway were utilized in a crude lysate mode, while the other enzymes were purified and used as single-added enzymes in the synthetic system. This is specified in the manuscript.

In Figure. 4, the enzymes in the precursor synthesis module (HemB, HemC, HemD and CobA) were used in the mode of purified enzymes, while the other Cob enzymes in the HBA pathway were used in a crude lysate. To clarify, we updated Figure. 4 to reflect this.

The systems using the HBA synthesis pathway were achieved by using HBA Cob enzymes crude lysate, meaning that all reactions in Figure 8a were a mixture of purified enzymes and crude lysate. On the other hand, the AdoCbl synthetic systems that utilized HBA as a substrate in Figure 8b were performed using purified enzymes without crude cell lysate.

- line 182 the reference to Fig 1a should be fig 3a?

- Thank you. Corrected.

- Fig. 4, What is the final concentration crude lysate? 10 g of HBA extract should be written as a concentration. Also, is that total protein or heterologously expressed protein?

- Thank you for pointing this out. We have added quantification of the HBA cell crude extract in the supplementary material, including the wet cell weight (WCW), dry cell weight (DCW) and

total protein (TP) per OD₆₀₀ of cell broth (figure S14). This was done because we quantified the HBA CCE using the OD₆₀₀ of cell broth during the experimental process. In the methods section, we wrote '10 g of HBA extract' to indicate the ratio of crude cell extract treatment, but we missed the volume in the last version of the manuscript. In the revised manuscript, we have standardized the usage of the HBA CCE by TP per volume of reactant in Table S2 and added a more detailed description in the text.

- line 264, the authors decide to modify buffer/pH but the rationale for testing this isn't clear. What was the authors hypothesis? Is there previous literature to suggest that this would be an important lever in the cell-free environment?

- We made modifications to the buffer and pH to ensure that the ion concentration and pH value of the enzymatic reaction did not affect the activity of the enzymes. To screen for a suitable buffer and pH value for a mixed multienzyme reaction, we evaluated the product titer and carried out tests by changing the buffer with varying pH values to identify the optimal reaction conditions.

Additionally, the reaction buffer and pH were previously optimized to increase the final product titer in the cell-free system (Wang W. et al. *Metabolic Engineering*. 2017. Meng D. et al. *ACS Catalysis*. 2020. Cheng K. et al. *Metabolic engineering*. 2018¹⁻³), and we have included these references in the manuscript for readers to better understand.

- Fig. 5, What pH is being referred to in this figure? Is this initial pH of the reaction, final pH, buffer pH before assembly of the enzymatic reaction? If this is of the buffer, how does this impact the reaction? Please clarify.

- The pH value shown in Figure 5 represents the initial pH of the reaction system. While enzymes generally have preferred pH ranges for optimal catalysis, these preferences can vary. Our objective was to identify a suitable pH condition that would accommodate all assembled enzyme reactions. To evaluate this, we analyzed the final titer of HBA under different reaction conditions. However, it is important to note that the identified pH value may differ when different enzymes are employed, as depicted in Figure 5b.

In Figure 5b, altering the pH value from 7.6 to 8.0 did not result in a significant improvement in HBA titer in the original reaction system (Ori). Nonetheless, the reaction system containing SAM module enzymes exhibited an almost one-fold increase in HBA production at pH 8.0 compared to pH 7.6. This finding suggests that the MetK and MtnN enzymes perform better at pH 8.0, thus enhancing HBA production.

- Fig. 6 is really small and difficult to read. The authors should consider reformatting to increase figure size. Maybe move d to be first panel because it is the first part referenced followed by current panel a on the first row above b and c, that would make a 3 row and 2 column figure allowing for all panels to be bigger and readable.

- We have restructured and enlarged Figure 6 to enhance its readability and better comprehension.

- Fig. 8, this is a great figure. However, I don't understand the term strengthen. The authors should be explicit about what is changing or the difference at each stage. I appreciate the conditions listed in table 2, but the term strengthen is used several times and has different meanings in each case. Please use more specific words for what is intended.

- In the earlier version of Figure 8, we presented a summary of the AdoCbl titer achieved using 5-ALA or HBA as substrates throughout the optimization process, emphasizing the significant improvements in titer. However, we have revised Figure 8 to enhance readability and comprehensibility of the optimization strategies. The reorganized figure now provides specific details about the changes and differences observed at each stage. Moreover, we have highlighted the synthetic strategies employed, distinguishing between a one-pot reaction system and a batch reaction system. Additionally, to provide a comprehensive understanding of each synthesis system, we have included Table S2, which offers detailed information to reviewers and readers.

Reviewer #2 (Remarks to the Author):

Overall Comments:

This paper describes an in vitro reconstitution of vitamin B12 biosynthesis from the early precursor 5-ALA, which is a complex process involving >30 enzymatic reactions. The authors document their optimization of the reaction conditions, as well as describing how they have overcome challenges in supplying reactants and avoiding feedback inhibition by incorporating a 'regeneration module' (to regenerate key reactants and consume potent inhibitors) to increase the yield. The strategies employed here will be generally useful for the development of other cell-free synthetic pathways. It remains unclear whether such a cell-free system is ever likely to compete with bacterial fermentation for production of vitamin B12 on a commercial scale.

The authors highlight several challenges associated with metabolic engineering to improve B12 synthesis (lines 54-56 and 560-564). However, these challenges seem minor relative to those which remain if vitamin B12 is to be produced on a commercial-scale by cell-free synthesis, with a potential for the text to mislead readers:

For example, line 561 states (in reference to engineered E. coli strains for B12 production) 'titers, yield and productivity remain discouragingly low, and further optimization is a challenge', but this seems at odds with the authors apparent optimism regarding cell-free synthesis since the yields per litre are no better. The latter also seems to be a technically more challenging (expensive) manufacturing route.

The authors also describe an E. coli strain which produces 0.67 mg/L from 24 hrs fermentation, which seems promising in comparison to the 0.4 mg/L of B12 reported here using cell-free

synthesis from an early precursor 5-ALA. Moreover, other engineered strains of E. coli have already generated a promising 2.9 mg/L (88 mg from 30 L culture) of an advanced intermediate HBAD, (Widner et al 2016, <https://doi.org/10.1002/anie.201603738>).

- **Compared to engineering industrial strains, a cell-free reaction platform offers several reported and discussed advantages.** For instance, in a cell-free system, the titer of monoterpenes has been achieved at 15 g/L, surpassing the limits of cellular toxicity that are difficult to overcome using cell-based systems⁴. Furthermore, the in vitro transformation of biomass into clean fuels, such as hydrogen (H₂), enables the simultaneous catalysis of glucose and xylose into H₂ with a maximum yield of two H₂ per carbon, which represents the highest possible yield⁵. Generally, a cell-free system offers advantages such as high titer, high yield, high tolerance to bio-toxicity, and ease of operation, which are challenging to achieve through cell engineering.

The optimization of vitamin B₁₂ production strains pose certain challenges. Industrial strains such as *Pseudomonas denitrificans*, *Propionibacterium*, and *Sinorhizobium meliloti* have reported titers ranging from 100 mg/L to 200 mg/L⁶⁻⁷. For a long time, these strains have limited engineering tools that can be employed (e.g., random mutagenesis and plasmid-based gene expression), long fermentation cycles, and require expensive and complex growth media. Our recent publication introduced a novel genetic tool (CRISPR-based) into *S. meliloti*, resulting in a 10%-25% increase in production. This was the beginning of introducing advanced genetic tools in metabolic engineering of industrial strains, but we still need to overcome many difficulties of engineering⁷.

On the other hand, engineered novel chassis cells like *E. coli* have reported vitamin B₁₂ titers ranging from 0.65 µg/g to 550 µg/g DCW⁸⁻¹⁰, corresponding to a maximum of 1.2 mg/L. Although *E. coli* benefits from a mature research background and established genetic engineering techniques that offer advantages in vitamin B₁₂ production, the long synthetic pathway, step-by-step genetic manipulation, and intricate relationship between strain growth and metabolic synthesis can impede progress in revolutionizing vitamin B₁₂ production using *E. coli*.

In contrast, our cell-free synthetic system for vitamin B₁₂, while still far from achieving the titers of industrial strains, surpasses the titer achieved by recombinant *E. coli* and demonstrates promising prospects for further progress.

In the development of our novel cell-free synthetic system for vitamin B₁₂, we leveraged the advantages of the cell-free approach to address various challenges. Firstly, we utilized the ability of the cell-free system to elucidate the catalytic processes of undefined enzymes, enabling us to accurately synthesize difficult-to-detect intermediates. This allowed us to establish a method for detecting these intermediates and identify specific reaction sites, thereby exploring the relationships between cascade reactions and uncovering scientific questions.

In terms of titer and productivity, the lower titer achieved when using 5-ALA as a substrate presented challenges for practical applications. Although this represents a significant scientific milestone, it is not suitable for industrial production. The vitamin B₁₂ synthetic pathway is highly

regulated and not fully understood, with reported enzyme K_m values in the micromole range, feedback inhibition susceptibility to by-products of low concentration, and unclear enzyme characteristics. Therefore, synthesizing vitamin B₁₂ de novo *in vitro* (from 5-ALA) is more of an experimental supplement and scientific exploration rather than a practical proposition for industrial applications. To explore potential industrial applications, we attempted to synthesize vitamin B₁₂ using HBA as a substrate, which reduced the difficulty of the synthetic pathway. This approach yielded a titer of 5.8 mg/L after 14 hours. Comparing the productivity of our cell-free synthetic system using HBA as a substrate, which is calculated at 0.41 mg/L/h, to that of industrial strains over an 8-day period (calculated at 1.04 mg/L/h with a final titer of 200 mg/L), we find that the difference is not significant. Additionally, the cell-free system offers the advantage of shortening the production period when compared to microbial fermentation processes.

In terms of the input-output ratio, the current cell-free system for vitamin B₁₂ does not possess clear advantages due to the high costs associated with enzyme production and the addition of cofactors. However, we are actively working to address these challenges. One approach is the utilization of enzyme immobilization, which is a mature technology, along with cofactor regeneration (as mentioned in the manuscript). Additionally, we are exploring novel combination reaction modes, such as synthesizing AdoCby *in vivo* and catalyzing AdoCby to vitamin B₁₂ using five enzymes *in vitro*. These approaches are currently being optimized and have already shown higher titers of vitamin B₁₂ compared to starting with HBA (although these specific results have not yet been reported).

Furthermore, the cell-free system holds immense potential for further progress. In our recently published article, we achieved a titer of nearly 30 mg/L of HBA *in vitro* by addressing the SAM-related issue using a different method than the one described in this manuscript¹¹. This titer surpassed what was achieved *in vivo*. In this manuscript, we also made adjustments to the subsequent synthetic reactions after HBA. Based on these results, we firmly believe that cell-free synthesis has significant potential for application in the production of vitamin B₁₂.

We have included this discussion in our revised manuscript to provide further details on the application of the cell-free approach for vitamin B₁₂ production.

-

Specific Comments:

Line 54: Prior work of Warren et al in engineering B12 pathways in E. coli should also be cited (eg Widner et al 2016, <https://doi.org/10.1002/anie.201603738>).

- The relevant references have been appropriately cited in the relevant sections.

Figure 1 is a very useful summary of the complete B12 synthesis process but the chemical structures are very small and chemical reactions very difficult to make out.

- Figure 1 has been redesigned to better illustrate the key chemical structures.

In Figure 1, it may be useful to label the branch modules 'Branch Module 1' (as it is) and 'Branch Module 2' (for reaction of 5,6-dimethylbenzimidazole to alpha-ribazole-5'-P) as it is not clear from first glance that these are two separate processes.

- Figure 1 has been revised accordingly.

Line 250 says 'no obvious increase in the precorrin-2 titer was observed after additional SAM supplementation'. This does not make sense since Fig S2b shows an increase in precorrin-2 until > 5 mM SAM is added. Please clarify/revise.

- The description has been revised.

Line 297: 'but CbiA were main catalyzed the HBA derivant cobyrrinate to cobyrrinate a,c-diamide in the natural anaerobiotic metabolic pathway'. The meaning of this sentence is unclear, please revise.

- The description has been revised.

Figure 4c: Suggest changing the x-axis legend to 'SAM or L-Met' (currently it looks like it refers to the ratio of SAM/L-Met).

- The x-axis has been changed to 'SAM or L-Met'.

Lines 426 and 429: 'Stoichiometrically limiting reagent' is more precise than 'stoichiometric limiting condition'

- The description has been changed accordingly.

Line 463: What is meant by 'stoichiometrically reduced the material waste'? Does this mean that there was no material waste? Please clarify.

- The previous description aimed to clarify that our regeneration system is capable of partially regenerating the substrate (5-ALA) and cofactors (NADH, ATP, and Gln), although complete salvaging of all components is not achieved. For example, the synthesis of one AdoCbl molecule necessitates the consumption of six L-Gln and results in the production of six L-Glu. However, in theory, only four ammoniums can be utilized to regenerate L-Glu, even though 0.75 L-Glu is used for NADH and 5-ALA regeneration. As a result, 1.25 L-Gln is still consumed. Nevertheless, the consumption of L-Gln has been reduced from 6 to 1.25, which is why we used the term 'stoichiometrically reduced the material waste'. However, to enhance clarity, we have revised the description in this manuscript to explicitly state that our system can only partially regenerate the substrate and cofactors used.

Line 475: What is meant by 'stoichiometrically reduce the demand for 5-ALA...'? Please clarify.

- The revised statement remains similar to the previous answer. Theoretically, the amount of

5-ALA required to synthesize one AdoCbl has been reduced from 8 to 7.25 through the regeneration of NADH and 5-ALA in the designed regeneration module. As a result, we used the term 'stoichiometrically reduce the demand for 5-ALA' in the previous manuscript. However, to enhance clarity, we have revised the description in this revised manuscript to explicitly state that our system can partially regenerate 5-ALA.

Line 482: Correct 'extraly introducing' with 'the additional introduction of'

- The description has been revised.

In Figure 8 'Productivity' is misspelt.

- The y axis title in Figure 8 has been corrected.

Line 518: There is insufficient data in Fig S9c to claim a 'linear dependence' on the quantity of strain. It would be acceptable to say 'the transformation yield is related to the quantity of strain'.

- The sentence has been modified to remove the 'linear dependence'.

Line 576: 'the cascade reaction was titrated, raising the positive driving force but the challenge of competitive relation in cascade reactions'. Meaning is unclear, please revise.

- The description has been revised.

The discussion section contains a summary of the work performed to optimize in vitro B12 synthesis but references to relevant data/figures are missing.

- We have relocated the optimization of the cell-free vitamin B₁₂ synthesis system to the Results section and linked it with the relevant data and figures.

Supplementary Table 1:

It should be made clear how delta G has been calculated for each transformation (ie at pH 8, ionic strength 0.1M or at pH 7.3, ionic strength 0.25M?) as well as which explicit chemical reactions this delta G corresponds to.

Footnotes are denoted 'a' and '1' – please make numbering consistent.

- The Table S1 has been revised accordingly. We added all reaction equations in this Table and information on calculating ΔG has been added in footnotes [a].

In Figure S2 it is not obvious how conditions differ between panels b-d. It would be helpful for key differences in conditions to be evident on the panels themselves (by including a '+MtnN' label on panel d, for instance).

- The subtitle of each panel has been added in the revised Figure S2.

Figure S4 is unclear.

-What does 'consumption' refer to in each panel? Ie in panel 'a' is this consumption of HBA? ATP?

-There are two y-axes on each panel but it is not clear which data corresponds to which y-axis.

-In Fig S4c what do triangles represent? It is not clear what the x-axis represents? What does 'proportion' refer to? Please clarify.

- Figure S4 (which is Figure S5 in the revised manuscript) has been completely reworked to enhance clarity. In the original figure, 'consumption' referred to the difference between the inputted HBA substrate and the total amount of detected HBA, HBAM, and HBAD, which corresponds to the amount of synthesized CBAD. To improve understanding, we have replaced 'consumption' with 'CBAD' to indicate the final product of the two-step cascade reaction. We have also modified Figure S4 to emphasize that the reaction from HBAD to CBAD is the driving force behind CBAD synthesis in this cascade reaction. While all three compounds (HBA, HBAM, and HBAD) were detected in the synthesis of HBAD from HBA, indicating incomplete catalysis, only a small amount of HBAM was observed in the synthesis of CBAD from HBA (7.60% of the total input), and the yield of CBAD was higher than that of HBAD (92.40% versus 36.53%). Additionally, we have included a simple diagram in the upper portion of Figure 5c to illustrate the driving force in this cascade reaction, with a bold arrow representing the strong driving force of CBAD synthesis.

Figure S8d: The right hand y-axis 'Titer/Biomass' – should the units be mg OD-1 (rather than mg-1 OD-1)?

- The y-axis in Figure S8d has been revised.

Figure S10b: It is very difficult to see the bacterial growth in the images of Salmonella bioassay plates. Is a better resolution picture available?

- We have updated Figure S10b (Figure S12b in the revised manuscript) with a higher resolution image of the results from the *Salmonella* bioassay.

Reviewer #3 (Remarks to the Author):

In the presented article "A synthetic cell-free 36-enzyme reaction system for vitamin B12 production" by Kang et. al., the authors describe an extensive and monumental effort to develop cell-free B12 vitamin production pathway and address major barriers for high titer/high productivity levels in vitro. This will be a significant addition to the field of synthetic biology and in vitro production of commodity chemicals as it demonstrates viability of cell free synthesis of exceptionally complex compound with input of diverse cofactors and efforts to recycle and regenerate these cofactors. While there was major work done to address the bottlenecks, to drive the flux of pathway downstream, the possible inhibition mechanisms and ways to address these issues, there are few revisions needed before this article is ready to be published. Overall, the work is substantial and adds to the toolkit of expanding field of cell-free biology.

1)HBA Module: why this module was processed via crude extract instead of purification of each enzyme? While it may be better to co-express all enzymes and utilize crude extract to synthesize products, the choice of lysate rather than pure enzymes seems to be a major pitfall of the project. The actual yield of HBA (and subsequent AdoCbl) is around 10% indicating that this module is the major bottleneck in B12 synthesis. The authors tried to address this low yield/titer by looking into feedback inhibition and reaction buffering. While SAM/SAH ratio plays a significant role in methyltransferase activity inhibition and authors cleverly overcome this obstacle with MetK/MtnN, the effect seems to be abolished past 3 mM SAM addition (Fig. S2). It seems that last 3 steps (CobK/CobL/CobH) are thermodynamically unfavorable in this module and the final equilibrium may be limiting the titer/yield. Authors need to address this possibility, maybe create a sink for HBA with introduction of RcCobB and see how the flux changes in this module.

- Initially, we constructed the HBA synthetic system using purified enzymes. However, we were disappointed to discover that a mixture of purified Cob enzymes was unable to produce HBA from 5-ALA, resulting in the detection of only Uro III as a dead-end by-product (Figure S1 a), while mixed crude cell extract of eight Cob enzymes from eight expressing strains could produce HBA in our previous work¹¹. We hypothesized that some of the Cob enzymes became unstable and lost their activity during the purification process. Through screening the Cob enzymes, we determined that only CobL and CobH remained active after common purification process, while the other Cob enzymes lost their catalytic activity. Considering the workload involved in optimizing the purification process for all six Cob enzymes, we decided to utilize a mixed crude cell extract (CCE) containing enzymes from the Cob operon to express all eight enzymes in a single strain. The use of the CCE of Cob enzymes significantly simplified the operation process and enabled the functionality of the cell-free reaction system.

Regarding the comment on MetK/MtnN, we acknowledge that the previous version of Figure S2 lacked subtitles, which may have caused confusion for reviewers and readers. Figure S2d demonstrates that the titer of Precorrin-2 decreases after the addition of 3 mM SAM in the reaction system due to the optimization of MtnN enzyme addition. However, the final optimization results with the addition of both MetK and MtnN are presented in Figure S2f, where the titer of Precorrin-2 does not significantly decrease when an excess methyl group donor (changed to L-Met instead of SAM in the final application) is added.

- To investigate the metabolic flux upon the introduction of the CobB enzyme for the conversion of HBA, we conducted additional experiments. However, we observed a slight decrease in the titer of HBA, and HBAD was hardly detected in each group after the introduction of CobB. We hypothesize that this result is attributed to the equilibrium of the CobB reaction, as discussed in the manuscript. Through our exploration of the cascade reaction involving CobB and CobNST, we have demonstrated that the driving force of the CobNST reaction can shift the equilibrium of the CobB reaction towards HBAD. Therefore, we conclude that introducing only the CobB enzyme is insufficient to drive the reaction equilibrium in the HBA reaction system. Further introduction of enzymes involved in subsequent reactions would aid in directing the metabolic flux towards the desired final product. We have included the supplementary result as Figure S3 in this revised manuscript.

Figure 1. Driving force of CobB reaction to HBA synthetic system

a, driving force of CobB reaction to HBA synthesis using ATP regeneration only. HBA: 0.1 μM HemB, 1 μM HemC, 1 μM HemD, 10 μM CobA, 5 μM MetK, 5 μM PpK, 10 μM MtnN, 2 mM AMP, 1 mM SMPP, 5 mM 5-ALA, 1 mM L-Met, 0.236 mM NADH, 0.236 mM NADPH, 2 mM MgCl_2 , 10 mM KCl, 5 mM NaCl, 155.8 mg/ml wet cell weight of HBA CCE (corresponding 17.81 mg/ml dry cell weight, 6.23 mg/ml total protein, and 10 OD_{600} HBA cell culture fluid per volume reactant) in 50 mM Tris-HCl (pH 8.0) buffer. HBAD1: HBA reactant with 5 μM CobB and 1 mM L-glutamine; HBAD2: HBA reactant with 10 μM CobB and 1 mM L-glutamine; HBAD3: HBA reactant with 15 μM CobB and 1 mM L-glutamine. b. driving force of CobB reaction to HBA synthesis using ATP and ATP regeneration both. HBA: 0.1 μM HemB, 1 μM HemC, 1 μM HemD, 10 μM CobA, 5 μM MetK, 5 μM PpK, 10 μM MtnN, 2 mM ATP, 1 mM SMPP, 5 mM 5-ALA, 1 mM L-Met, 0.236 mM NADH, 0.236 mM NADPH, 2 mM MgCl_2 , 10 mM KCl, 5 mM NaCl, 155.8 mg/ml wet cell weight of HBA CCE (corresponding 17.81 mg/ml dry cell weight, 6.23 mg/ml total protein, and 10 OD_{600} HBA cell culture fluid per volume reactant) in 50 mM Tris-HCl (pH 8.0) buffer. HBAD1: HBA reactant with 5 μM CobB and 1 mM L-glutamine; HBAD2: HBA reactant with 10 μM CobB and 1 mM L-glutamine; HBAD3: HBA reactant with 15 μM CobB and 1 mM L-glutamine.

2) These pot reactions are expected to be acidic over time (based on by-products such as CO_2) and authors changed the buffer to HEPES-NaOH to address this matter. While it may not be obvious initially, these incremental drops in pH destabilize the system and inactivate enzymes thereby lowering the titer and yields. Recent work in NC on cell-free isobutanol production (<https://doi.org/10.1038/s41467-020-18124-1>) showed that control of optimal pH in reactions considerably increased the titers and productivity of overall system. Authors need to clarify and address this obvious barrier that may significantly alter their overall system. In the future, they may also test better buffer systems (such as bicarbonate solutions).

- To verify and address the pH changes in our reaction system, we chose to monitor and adjust the

HBA synthetic system instead of the AdoCbl system for several reasons. Firstly, there are two reactions in the HBA synthetic pathway that can produce CO₂: one catalyzed by CobL and another in the threonine consumption branch pathway (referred to as the threonine pathway below) catalyzed by CobC. Secondly, we performed the threonine pathway reaction separately in the higher-production AdoCbl reaction system (No. 6, 7, 8 reaction groups in Figure 8a and No. 2, 3, 4 reaction groups in Figure 8b). Therefore, any CO₂-induced pH change in this reaction would not affect the AdoCbl synthetic system, as the final AdoCbl synthetic reaction would adjust the pH after CO₂ has already been synthesized. Lastly, the HBA system exhibited a higher titer than the AdoCbl system, enabling a stronger demonstration of the pH changes in our reaction.

Figure 2 in this report illustrates the pH and titer changes during HBA synthesis. In the Unregulated reaction group, where no pH adjustment was made, the pH value decreased from 7.85 at the start of the reaction (1 h) to 7.07 at 12 h, and finally to 6.52 at 24 h. However, in the Adjusted reaction group, where we maintained the pH between 7.67 to 8.01 for the first 12 h and allowed it to self-adjust in the latter 12 h, the pH only decreased to 7.25 at 24 h. The HBA titer in the Ori group reached 8.54 mg/L at 12 h and 8.64 mg/L at 24 h, while the Adjusted group achieved 8.66 mg/L at 12 h and 10.58 mg/L at 24 h.

These results indicate that CO₂ accumulation can lead to pH alterations in our cell-free system, which can have a slight impact on the final production. However, controlling the optimal pH during reactions demonstrates the potential to enhance the final titer. In addition, we have included supplementary results and discussion in this revised manuscript to provide further insights and information.

Figure 2. pH monitoring in HBA synthesis and influence of controlling pH in HBA synthesis
a, driving force of CobB reaction to HBA synthesis using ATP regeneration only. HBA: 0.1 μM HemB, 1 μM HemC, 1 μM HemD, 10 μM CobA, 5 μM MetK, 5 μM PpK, 10 μM MtnN, 2 mM AMP, 1 mM SMPP, 5 mM 5-ALA, 1 mM L-Met, 0.236 mM NADH, 0.236 mM NADPH, 2 mM MgCl₂, 10 mM KCl, 5 mM NaCl, 155.8 mg/ml wet cell weight of HBA CCE (corresponding 17.81 mg/ml dry cell weight, 6.23 mg/ml total protein, and 10 OD₆₀₀ HBA cell culture fluid per volume reactant) in 50 mM Tris-HCl (pH 8.0) buffer. HBAD1: HBA reactant with 5 μM CobB and 1 mM L-glutamine; HBAD2: HBA reactant with 10 μM CobB and 1 mM L-glutamine;

HBAD3: HBA reactant with 15 μM CobB and 1 mM L-glutamine. b. driving force of CobB reaction to HBA synthesis using ATP and ATP regeneration both. HBA: 0.1 μM HemB, 1 μM HemC, 1 μM HemD, 10 μM CobA, 5 μM MetK, 5 μM PpK, 10 μM MtnN, 2 mM ATP, 1 mM SMPP, 5 mM 5-ALA, 1 mM L-Met, 0.236 mM NADH, 0.236 mM NADPH, 2 mM MgCl_2 , 10 mM KCl, 5 mM NaCl, 155.8 mg/ml wet cell weight of HBA CCE (corresponding 17.81 mg/ml dry cell weight, 6.23 mg/ml total protein, and 10 OD_{600} HBA cell culture fluid per volume reactant) in 50 mM Tris-HCl (pH 8.0) buffer. HBAD1: HBA reactant with 5 μM CobB and 1 mM L-glutamine; HBAD2: HBA reactant with 10 μM CobB and 1 mM L-glutamine; HBAD3: HBA reactant with 15 μM CobB and 1 mM L-glutamine.

3)Line 339 states that excessive ATP negatively affected the product titers which is surprising. Is there any particular reason for this observation? The ATP regeneration system seems to have no such adverse effect, probably due to titrating and optimizing PpK levels. If it is so, authors need to state these findings.

- During our experimental process, we observed that adding an excessive amount of ATP to the reactant led to protein precipitation. This phenomenon is hypothesized to occur due to the precipitation of enzymes upon the introduction of ATP, potentially triggered by the slight pH change immediately after adding ATP. Fresh ATP solution tends to be acidic, while fresh AMP solution is alkaline. Adding a high concentration of ATP solution directly to the protein solution resulted in partial protein precipitation due to the resulting pH shift. However, adding AMP solution did not lead to protein precipitation. When measuring the pH value of the reactant after varying amounts of ATP were added, we observed no significant change, indicating that the instantaneous change in enzyme activity was not readily detectable.

Nonetheless, when we replaced the direct input of ATP with an ATP regeneration system utilizing AMP, we observed no significant protein precipitation in the reactant. Therefore, we concluded that excessive ATP can cause protein precipitation, leading to alterations in enzymatic activity and a negative impact on the cascade reaction. However, the use of an ATP regeneration system can alleviate this issue.

We have incorporated the aforementioned results and hypotheses into the revised manuscript.

Figure 3. Titrating ATP input in CobB-CobNST cascade reaction.

The composition of reactions above is same as ones described in manuscript, but the input of ATP was titrated from 1 to 20 mM.

4) While regeneration system overall is elegant and increases the titers of overall system by ~16%, it is not stoichiometrically balanced and it seems to have more supplemental role rather than fully recycling/regenerating system. Moreover, the regeneration system may be compromised and needs to be optimized further with the use of crude extract as lysates contain residual non-specific ATPase and NADH/NADPH oxidase activities that may alter the balance in the reaction. It is not clear whether increase in titer is due to overall module or parts of the system such as PpA and MetK/MtnN.

- Our regeneration system is not stoichiometrically balanced but serves a supplementary role in supporting our system and increasing production titers. Most of the regenerating enzymes used in this article were individually tested, such as RocG for regenerating NADH (Figure S8a), GlnA for regenerating L-Gln (Figure S8b), and PpK for regenerating ATP, as well as PpA for decomposing triphosphate (Figure S8c). In the revised manuscript, we have provided more conclusive information regarding the titer improvements achieved by implementing the regeneration system. For instance, we observed a 3.3-fold increase in HBA production with the SAM module, a 3.0-fold increase in AdoCby (synthesized from HBA) with PpK, RocG, and GlnA, a one-fold increase in AdoCbl (synthesized from 5-ALA) in a one-pot synthetic system, and a 16.7% increase in AdoCbl (synthesized from 5-ALA) with regeneration module 1 in batch reaction systems.

In summary, we utilized the regeneration system both partially and integrally at different stages, and all components contributed to the production of AdoCbl or intermediates. The use of crude cell extract may affect the regeneration enzymes due to the presence of residual endogenous enzymes in the 5-ALA-initiated system. However, this negative effect was minimized in the HBA-initiated system by excluding the crude cell extract in later pathway reaction designs. The

impact of residual endogenous enzymes on regeneration will be taken into consideration for future optimizations.

5) 5.78 mg/L were obtained after full assembly of the system but it is not clear if the pathway is truly one pot or the final product is synthesized via modules, intermediates are extracted to be further processed (such as AdoCBAD and AdoCby synthesis in anaerobic conditions). The volume of each reaction is not clear from methods sections and whether these findings would scale on a higher volume. It would be beneficial to have a table of final concentration of each enzyme used in the final assembly.

- Our revised Figure 8 provides a clear depiction of which reactions were conducted in one-pot mode or batch mode. Additionally, we have outlined the operating conditions for the batch method and specified the final concentration of each enzyme in Table S2. We have also included the final reaction volume in Table S2, ranging from 0.25 ml to 5 ml. However, it is important to note that we have not yet scaled up our reaction to a larger volume.

Minor Revisions:

Line 40: change to 5'-deoxyadenosine

- Revised.

Line 213-214: CysG is at the end of operon and it is not clear what CysGA and CysGB refer to. That would help to explain fig. 4A better.

- We have added references for the CysG enzyme in this section, and we have also included a simple illustration in Figure 4a to help readers better understand.

Line 293-305: while the authors performed genome mining for AdoCby module, they chose S. meliloti HemBCD genes in precursor module. Is there any reason/discussion for this approach? Given the immense size of the project, was genome mining done in an ad hoc fashion?

- Based on previous research conducted in our laboratory regarding the reconstruction of the vitamin B₁₂ synthetic pathway in *E. coli*^{9-10, 12}, some enzyme mining had already been performed through *in vivo* experiments. As a result, we did not attempt to mine all of the enzymes in the synthetic pathway due to the substantial workload involved. However, during the initial stages of both *in vivo* fermentation and *in vitro* reactions, we observed significant accumulation of large amounts of HBA in the reactant. To address this intermediate accumulation, we conducted additional enzyme mining and gained insights into the driving force behind the CobB and CobNST cascade reactions.

Initially, when establishing this cell-free reaction platform, we did not extensively engage in enzyme mining work. This decision was made to strike a balance between our workload and experimental goals. In this manuscript, enzyme mining was carried out based on the identification of bottleneck reactions in the reactant. However, in the early stages, it was challenging to identify blocked sites due to the limited availability of methods for detecting most intermediates. Nevertheless, with the establishment of a detection method for AdoCby and AdoCBAD in this manuscript, we anticipate that we will be able to identify more bottleneck reactions and conduct

further enzyme mining to address them.

Line 482-482: Needs grammatical revision.

- Revised.

Line 531-532: Fig.8 labels: change the label to “productivity”

- Revised.

Line 966-967. Needs grammatical revision

- Revised.

Reviewer #4 (Remarks to the Author):

Kang and collaborators assembled a pathway for in vitro biosynthesis of vitamin B12 (adenosylcobalamin, AdoCbl). Authors performed a cascade of catalytic reactions that start from 5-aminolevulinic acid (5-ALA) as an inexpensive substrate. More than 32 biocatalytic reactions were integrated and optimized to achieve complete cell-free synthesis of AdoCbl, after overcoming feed-back inhibition, analytical issues, instability of intermediates, and imbalance and competition of cofactors. Authors report that this cell-free system produced 417.41 µg/L and 5.78 mg/L of AdoCbl using 5-ALA and the purified intermediate product hydrogenobyrate as substrates, respectively. This study is interesting from a technical point of view (36 enzymes catalyzing 30/32 reactions), but neither the product nor the strategy are novel/interesting. All the enzymes have been extensively described in literature. All the biochemical steps have been thoroughly characterized by other researchers. Mixing the enzymes together and measuring product formation does not come across as particularly disruptive. The regeneration system used by the authors is not novel, either, and has been used multiple times to reconstitute cofactors in cell-free systems (in example, ATP). Worryingly, the authors do not even know all the chemical transformations that happen in their system (Lines 514-518). In summary, while the technical feat of assembling a complex pathway has some interest, the level of novelty and impact in the biocatalysis community is limited. A more specialized journal (ACS Catalysis, Metabolic Engineering) is recommended as a venue for this study.

- The vitamin B₁₂ synthesis pathway is an ancient, finely regulated, and delicate process that generates one of the most structurally complex small molecules found in nature, weighing over 1500 Daltons, through sequential modifications. Reconstructing the vitamin B₁₂ synthetic pathway is challenging both *in vivo* and *in vitro* due to the involvement of numerous catalytic enzymes, susceptibility to feedback inhibition in multiple reactions, requirement for diverse cofactors, and lack of detection methods for intermediates within the pathway. To overcome these challenges, we divided the entire pathway into modules, optimized each module by introducing additional enzymes to reduce feedback inhibition and regenerate cofactors, and developed novel detection methods for non-commercial intermediates.

In addition to addressing technical challenges, we also tackled scientific questions, such as uncovering the driving force behind certain cascade reactions in the pathway and identifying the

cofactor reductase activity of CobR. Regarding the cofactor regeneration system, we utilized the PpK enzyme to regenerate ATP based on previously reported multienzyme synthesis studies. Additionally, we employed regeneration module 1 to replenish both NADH and 5-ALA, specifically designed to address the unique byproducts present in our synthetic pathway. Furthermore, we explored the regeneration of cofactors by utilizing byproducts such as L-Gln and ammonium, rather than introducing additional consumed compounds (e.g., methanol) to regenerate NADH. This approach aimed to prevent the accumulation of byproducts that could potentially disrupt our reaction system. It represents a creative design strategy to regenerate cofactors and reduce the accumulation of byproducts in the assembly of a lengthy and multifaceted cell-free synthetic pathway *in vitro*.

In addition to our efforts to synthesize vitamin B₁₂ *de novo* from 5-ALA, we also investigated the potential industrial applications of cell-free synthesis using HBA as a substrate. Interestingly, using HBA as a starting point resulted in a higher final titer of the product compared to starting with 5-ALA. This finding opens up the possibility of utilizing cell-free systems to establish a novel approach for vitamin B₁₂ production, combining *in vivo* and *in vitro* synthesis or cell-free synthesis.

To summarize, our manuscript encompasses scientific exploration of uncertainties within the vitamin B₁₂ synthetic pathway, provides guidance for assembling a complex and extensive cell-free system, and presents a preliminary evaluation of this innovative production mode for vitamin B₁₂. This research will be of significant interest to experts across various fields, including biology, biotechnology, food, and nutrition.

‘Worryingly, the authors do not even know all the chemical transformations that happen in their system (Lines 514-518)’,

The uncertain reaction did not occur within our reaction system or reaction pathway but rather during the post-processing phase of our detected samples. To address this issue and ensure accurate results, we developed a safe post-processing method to stabilize the photosensitive AdoCbl without using cyanide. In order to clarify and provide a comprehensive understanding of our findings, we included the direct detection results of our synthetic AdoCbl (figure S10 a) and proposed hypotheses regarding the reactions occurring in this novel post-processing method. Furthermore, we expanded the application of this post-processing method to stabilize the intermediate AdoCby (figure S11), demonstrating its universality to AdoCbl and its derivatives. These additional details have been incorporated into our revised manuscript to enhance the clarity and validity of our results.

Other comments:

+ Urogen should be synthesized under aerobic conditions. The cascade catalysis mentioned by the authors is not clear for this Reviewer. What is the stability of uroporphyrinogen? This should be reported.

- We synthesized uroporphyrinogen III in an anaerobic chamber through the catalysis of HemB, HemC, and HemD enzymes using 5-ALA as a substrate. Subsequently, we rapidly oxidized it to uroporphyrin III using a boiling water bath. The auto-oxidation of uroporphyrinogen III into

uroporphyrin III at 30°C was monitored every hour and is illustrated in Figure S1b. We observed that uroporphyrinogen III is unstable, as evidenced by 83.3% of uroporphyrinogen III being oxidized after 7 hours of exposure to air at 30°C. In this revised manuscript, we have included the mass spectrometry (MS) detection results of uroporphyrinogen III and uroporphyrin III within the inner frame of Figure S1b to provide a clear description of the reactions occurring during the preparation process of these two compounds.

+ *L211: Large (???) plasmid?*

- Revised

+ *All figures in the main text lack statistical comparisons, making the optimization of steps claimed by the authors dubious.*

- The statistical comparison was added in experimental groups that need to be compared with each other.

+ *The mass relationships for all the enzymes used in the cascade is not explained. How did authors optimize this parameter (how much enzyme is added for each component)?*

- The content of enzymes used in every system was listed in Table S2. Due to the large number of enzymes involved in the final synthetic system, we did not individually titrate every enzyme in the AdoCbl synthetic system. The utilization of enzymes commenced with the initial assembly of the test system in modules and gradually increased in the optimized systems to enhance the corresponding reactions.

1. Wang, W.; Liu, M.; You, C.; Li, Z.; Zhang, Y. P., ATP-free biosynthesis of a high-energy phosphate metabolite fructose 1,6-diphosphate by in vitro metabolic engineering. *Metabolic engineering* **2017**, *42*, 168-174.
2. Meng, D.; Wei, X.; Bai, X.; Zhou, W.; You, C., Artificial in Vitro Synthetic Enzymatic Biosystem for the One-Pot Sustainable Biomanufacturing of Glucosamine from Starch and Inorganic Ammonia. *ACS Catalysis* **2020**, *10* (23), 13809-13819.
3. Cheng, K.; Zheng, W.; Chen, H.; Zhang, Y. P. J., Upgrade of wood sugar d-xylose to a value-added nutraceutical by in vitro metabolic engineering. *Metabolic engineering* **2019**, *52*, 1-8.
4. Korman, T. P.; Opgenorth, P. H.; Bowie, J. U., A synthetic biochemistry platform for cell free production of monoterpenes from glucose. *Nature communications* **2017**, *8*, 15526.
5. Rollin, J. A.; Martin del Campo, J.; Myung, S.; Sun, F.; You, C.; Bakovic, A.; Castro, R.; Chandrayan, S. K.; Wu, C. H.; Adams, M. W.; Senger, R. S.; Zhang, Y. H., High-yield hydrogen production from biomass by in vitro metabolic engineering: Mixed sugars coutilization and kinetic modeling. *Proceedings of the National Academy of Sciences of the United States of America* **2015**, *112* (16), 4964-9.
6. Noh, M. H.; Lim, H. G.; Moon, D.; Park, S.; Jung, G. Y., Auxotrophic Selection Strategy for Improved Production of Coenzyme B12 in Escherichia coli. *iScience* **2020**, *23* (3), 100890.
7. Cui, Y.; Dong, H.; Tong, B.; Wang, H.; Chen, X.; Liu, G.; Zhang, D., A versatile Cas12k-based genetic engineering toolkit (C12KGET) for metabolic engineering in genetic

manipulation-deprived strains. *Nucleic acids research* **2022**, *50* (15), 8961-8973.

8. Ko, Y.; Ashok, S.; Ainala, S. K.; Sankaranarayanan, M.; Chun, A. Y.; Jung, G. Y.; Park, S., Coenzyme B12 can be produced by engineered *Escherichia coli* under both anaerobic and aerobic conditions. *Biotechnol J* **2014**, *9* (12), 1526-35.

9. Fang, H.; Li, D.; Kang, J.; Jiang, P.; Sun, J.; Zhang, D., Metabolic engineering of *Escherichia coli* for de novo biosynthesis of vitamin B12. *Nature communications* **2018**, *9* (1).

10. Li, D.; Fang, H.; Gai, Y.; Zhao, J.; Jiang, P.; Wang, L.; Wei, Q.; Yu, D.; Zhang, D., Metabolic engineering and optimization of the fermentation medium for vitamin B(12) production in *Escherichia coli*. *Bioprocess and biosystems engineering* **2020**, *43* (10), 1735-1745.

11. Xiao, K.; Kang, Q.; Xiang, M.; Gong, D.; Fang, H.; Tu, X.; Zhang, D., Optimization of Hydrogenobyric Acid Synthesis in a Cell-Free Multienzyme Reaction by Novel S-Adenosyl-methionine Regeneration. *ACS synthetic biology* **2023**, *12* (4), 1339-1348.

12. Jiang, P.; Fang, H.; Zhao, J.; Dong, H.; Jin, Z.; Zhang, D., Optimization of hydrogenobyric acid biosynthesis in *Escherichia coli* using multi-level metabolic engineering strategies. *Microb Cell Fact* **2020**, *19* (1), 118.

REVIEWERS' COMMENTS

Reviewer #1 (Remarks to the Author):

The authors have sufficiently addressed all of my comments.

Reviewer #2 (Remarks to the Author):

It is pleasing to see that in the revised manuscript the authors have added to the discussion section suitable caveats about the potential commercial use of cell free synthesis for the production of vitamin B12 (including a discussion of innovations still needed for this method to potentially become an industrial process).

However, the abstract does still give the impression that this method is already a credible alternative for industrial B12 manufacturing (which it is not yet). I suggest that authors remove the work 'alternative' from line 25 so that it reads: 'Here, we report a method for the synthesis of AdoCbl based on a novel cell-free reaction system'.

Reviewer #3 (Remarks to the Author):

Dear Dr. Zhang,

Thank you very much for addressing the concerns I raised and I am recommending your revised article to be printed as presented. Your detailed explanations gave more in-depth analysis of the pathway that I missed in the initial review and given the complexity of the system, you and your research group have done outstanding job in addressing each module and productivity of entire pathway. The research is noteworthy in the field of cell-free biosynthesis and the experimental approach is novel and meticulous. I am looking forward for more publications from your research group.

Reply to Reviewers' comments point-by-point:

Reviewer #1 (Remarks to the Author):

The authors have sufficiently addressed all of my comments.

- Thank you for your valuable feedback and constructive criticism during the peer review process.

Reviewer #2 (Remarks to the Author):

It is pleasing to see that in the revised manuscript the authors have added to the discussion section suitable caveats about the potential commercial use of cell free synthesis for the production of vitamin B12 (including a discussion of innovations still needed for this method to potentially become an industrial process).

However, the abstract does still give the impression that this method is already a credible alternative for industrial B12 manufacturing (which it is not yet). I suggest that authors remove the work 'alternative' from line 25 so that it reads: 'Here, we report a method for the synthesis of AdoCbl based on a novel cell-free reaction system'.

- We revised the abstract and deleted the 'alternative' in line 25. Thank you for your valuable feedback and constructive criticism during the peer review process.

Reviewer #3 (Remarks to the Author):

Dear Dr. Zhang,

Thank you very much for addressing the concerns I raised and I am recommending your revised article to be printed as presented. Your detailed explanations gave more in-depth analysis of the pathway that I missed in the initial review and given the complexity of the system, you and your research group have done outstanding job in addressing each module and productivity of entire pathway. The research is noteworthy in the field of cell-free biosynthesis and the experimental approach is novel and meticulous. I am looking forward for more publications from your research group.

- Thank you for your valuable feedback and constructive criticism during the peer review process.